# Technical note on long-term probe misalignment and proposed quality control using the heat pulse method for transpiration estimations

Elisabeth K. Larsen[1,3], Jose Luis Palau[1], Jose Antonio Valiente[1], Esteban Chirino[2] and Juan Bellot[3,4].

[1]Mediterranean Centre for Environmental Studies (Fundacion CEAM), Charles R. Darwin 14, Parque Tecnològico, 46980 Paterna, Valencia, Spain
[2]Faculty of Agricultural Sciences, Lay University Eloy Alfaro of Manabí, Ave. Circunvalación, Vía San Mateo, Manta 130802, Ecuador
[3]Department of Ecology, University of Alicante, Apdo. 99, E-03080 Alicante, Spain
[4]IMEM Ramón Margalef, Department of Ecology, Faculty of Sciences, University of Alicante, 03080 Alicante, Spain

*Correspondence to* Elisabeth K. Larsen (eli@ceam.es)

## 1 Abstract

Transpiration is a crucial component in the hydrological cycle and a key parameter in many disciplines like agriculture, forestry, ecology and hydrology. Sap flow measurements are one of the most widely used approaches to estimate whole-plant transpiration in woody species due to its applicability in different environments and in a variety of species, as well as having the capacity of continuous high temporal resolution measurements. Several techniques have been developed and tested under different climatic conditions and in different wood properties. However, the scientific literature also identifies considerable sources of error when using sap flow measurements that need to be accounted for: probe misalignment, wounding, thermal diffusivity and stem water content. This study aims to integrate probe misalignment as a function of time to improve measurements during long-term field campaigns (> 3 months). The heat ratio method (HRM) was chosen because it can assess low and reverse flows. Sensors were installed in four individuals of *Pinus halepensis* for 20 months. Pines were located in a coastal valley in South-Eastern Spain (39º57'45" N 1º8'31" W) characterized by a Mediterranean climate. We conclude that even small geometrical misalignments in the probe placement can create a significant error in sap flow estimations. Additionally, we propose the need for new statistical information to be recorded during the measurement period, which can be used as a quality control of the sensor output. Relative standard deviation and slope against time of the averaged $\frac{v_1}{v_2}$ were used as quality indicators. We conclude that no general time limit can be decided for the longevity of the sensors but should rather be determined from individual performance over time.

## 2 Introduction

Plant transpiration is a key process in the hydrological cycle, generally being the largest component of total evapotranspiration in forest ecosystems (Schlesinger & Jasechko et al., 2014). However, accurate estimations of transpiration are still difficult to obtain, making field assessments of transpiration estimations crucial in hydrological planning, as well as in forestry, ecophysiological research and climate forecasting. Sap flow measurements are one of the most widely used approaches to estimate transpiration in woody species as it is readily automated for continuous readings and not limited to single leaf measurements (Forster, 2017; Peters et al., 2018;

Flo et al., 2019). Although some sap is stored in the stem and leaves, the majority (99%), is lost through transpiration. Sap flow measurements can therefore be used to directly estimate transpiration values (Forster, 2017).

There are a range of techniques used to measure sap flow, with method-families varying between heat dissipation (HD) (Granier, 1987; Lu et al., 2004), stem heat balance (SHB) (Langensiepen et al., 2014), trunk segment heat balance (THB) (Smith and Allen, 1996), and heat pulse velocity (HPV) (Marshall, 1958). However, they are all based on tracing heat within the xylem (Burgess et al., 2001; Davis et al., 2012; Forster, 2017). Marshall (1958) developed a theoretical method to determine sap flow from thermal diffusion and dissipation theory of heat pulses in wood. His technique relies on calculating the heat ratios measured in two parallel thermocouples aligned vertically and symmetrically with respect to a line heater along the direction of the xylem (Fig. 1). Burgess et al. (2001) developed an improved HPV technique, termed 'the heat ratio method' (HRM), based on the Marshall (1958) methodology. The HRM is sensitive to the direction of the sap flow, being able to measure low and reverse rates (Burgess et al., 2001). The HRM is therefore more appropriate to use in water-deficit environments were flow rates are low. Burgess et al. (2001) developed two steps of corrections for sap flow calculations by considering probe misalignment and wounding, caused by the implementation of the sensors (Fig. 1). By accounting for these sources of error and additionally estimating the stem moisture content and radial variability, the HRM has been evaluated as the HPV-method with the highest accuracy, although with a tendency of underestimating transpiration values (Forster, 2017), an error that is shown to increase with higher sap flow rates (Steppe et al., 2010).

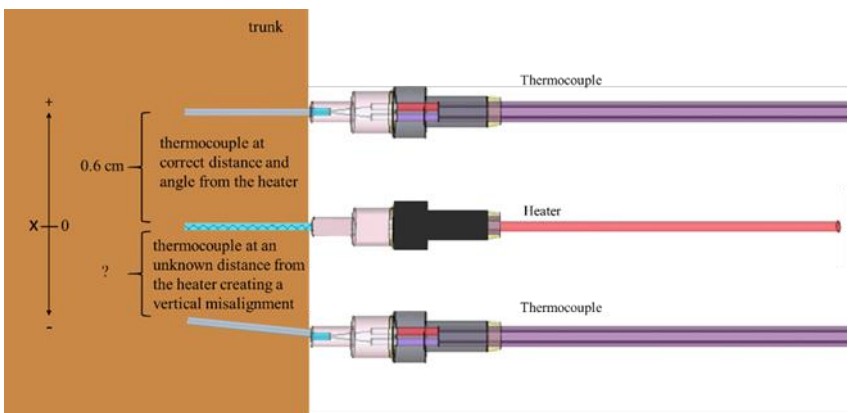

**Figure 1.** Schematic drawing of the three probes used for the heat ratio method: two thermocouples and one heater. One thermocouple demonstrates the correct placement, aligned vertically and symmetrically at 0.6 cm from the heater. The other thermocouple demonstrates the misalignment: when the probe is closer to the heater and not aligned symmetrically. The zero is marked at the height of the heater, indicating that the placement of the thermocouples are estimated in relation to the heater, at 0.6 cm and -0.6 cm distance, as indicated by the arrows. Sensors are modified from Davis et al. (2012) with permission.

Previous studies have suggested additional solutions for probe misalignment (Ren et al., 2017), for determining thermal diffusivity (Vandegehuchte and Steppe, 2012), and correcting for heterogeneous heat capacity in wood (Becker and Edwards, 1999). However, there is no recent recommendation for how long newly deployed sap flow sensors can be employed. Some studies have shown how sensor probes inserted into the xylem can dampen the signal due to blocking or destruction of vessels (Moore et al., 2010; Wiedemann et al., 2016). One way to avoid changes over time has been to reinstall sensors throughout the study period (Moore et al. 2010), interrupting continuous measurements in long-term datasets (Moore et al. 2010). Furthermore, there is little information to be

found on the exact interval for which sensors should be reinstalled (Vandegehuchte and Steppe, 2013). Therefore, we aim to find a quality indication that can ensure that readings do not deteriorate over time, or if they do, that it would be detectable. Attention should be given to check the accuracy of the raw readings, on which the rest of the methodology is built on. In addition, to allow for sensors to be employed over longer periods, it is necessary to develop a dynamic probe misalignment correction method due to observed changes in probe position over time. The adjustment proposed here, which is built on the calculations of Burgess et al. (2001), is necessary when monitoring transpiration continuously for more than 3 months due to an observed shift in probe placement from one season to another (Fig. 4, 6). Wood properties, the heterogeneity of the xylem, and plant tissue growth might further misplace the sensor after its implementation (Barrett et al., 1995). Burgess et al. (2001) corrected for linear probe misalignment *in situ* when an enforced zero sap flow was attained by severing root systems at the end of the experiment. However, this solution is not suitable for long-term measurements, as the misplacement will change over time (Ren et al., 2017), or when intrusive methods are not an option. Thus, the objective of this research is firstly, to develop a statistical filtering method to ensure the quality and consistency of the measurements over time, and secondly, to implement a modified version of the probe-placement calculation for sap flow series longer than 3 months.

This technical note is structured in two parts: the first one, dealing with the statistical analysis of long-term time series of averaged heat ratios to ensure the quality of data and their stability over time; the second part, proposing an adaptation of the method developed by Burgess et al. (2001) to introduce a dynamic probe misalignment correction for the HRM. The aim is to obtain a more precise calculation of transpiration by parameterising the probe misalignment as a function of time to correct for the effect of tree growth.

## 3    Materials and method

### 3.1  Field site

This study was carried out at an inland experimental plot located in the Turia river basin, Eastern Spain (39º57'45" N 1º8'31" W), in a Mediterranean climate with continental features. Average annual rainfall is 475 mm year $^{-1}$, average annual maximum and minimum temperature is 15.5 ºC and 4.4 ºC respectively. Sap flow sensors were installed in four pine trees (*Pinus halepensis* Mill.) according to the heat ratio method (HRM, Burgess et al., 2001). Each sap flow sensor consists of three needles: one heater and two thermocouples. We will refer to the thermocouples as probes, and when using the term "sensors" we refer to both probes and the heater. To establish a criterion that unified the effect of the hillslope, all sensors were drilled into the uphill side of each tree trunk. It was assumed to offer a greater consistency regarding soil water retention, possibly higher at this orientation. All sensors were covered with radiation shields. Since *P. halepensis* has a higher sap velocity near the cambium with the velocity steady declining nearer to the heartwood (Cohen et al., 2008), sensors were installed at 20 mm depth below the cambium to account for the average sap velocity rate, as estimated by Manrique-Alba (2017).  A metal plate was used as a guide during installation to ensure 0.6 cm spacing between drilled holes. The selected pines had an average diameter of 24.5 cm at breast height. Continuous measurements were obtained from April 2017 to December 2018.

### 3.2 Environmental conditions

Air relative humidity (%) and air temperature (ºC) were registered every 30-minute (U23 Pro V2, Onset Computer Corporation, USA). Precipitation was registered using a rain gauge with 0.2 mm resolution (RGR-M, Onset Computer, USA). Three soil moisture probes were inserted at 20-25 cm depth to register soil water content, SWC, (S-SND.M005, Onset Computer Corporation, USA), using a datalogger for logging specifications (HOBO Micro Station, USA). To assume periods of zero sap flow, relative extractable water (REW), was calculated using the method of Bréda et al. (1995):

$$REW = \frac{(\theta_t - \theta_{min})}{(\theta_{max} - \theta_{min})}$$

[1]

Where $\theta_t$ is the registered SWC, $\theta_{min}$ is the minimum SWC observed during the measurement period, and $\theta_{max}$ is the SWC at field capacity. Values surpassing one were converted into one, according to Granier et al. (2000).

### 3.3 Construction of the sensors

The thermocouples were made after Davis et al. (2012), with a type E junction of chromium and constantan. The wires of the thermocouples were soldered together at temperatures not surpassing 200°C, placed 20 mm inside a micropipette (10μL), and then into a needle (0.12 cm x 4 cm, Sterican, Braun). The heater was made from a constantan wire of 20 mm, coiled around a 7 cm long aluminum wire and placed inside the same type of needle as the thermocouples. The wire-ends were then soldered on to an electrical cable. A resistance of 4.95 ohms was then soldered onto to the cable to get a total electrical resistance of 14.95 ohms. The heater was connected through a solid-state relay to a 12 V battery, delivering 7.0 W of power when emitting heat pulses. Both thermocouples were connected to a CR800 datalogger for measurement recording (Campbell Scientific Inc., USA).

### 3.4 Quality control

Marshall (1958) parameterised the instantaneous heat pulse velocity (V) in the HRM as a function of time (s) following a heat pulse, and an instantaneous heat pulse ratio (HPR), defined as HPR $= \frac{v_1}{v_2}$, where $v_1$ and $v_2$ are the downstream and upstream temperature increases, respectively, measured after the release of a heat pulse:

$$V = \frac{4Kt \ln\left(\frac{v_1}{v_2}\right) - (x_2^2 + y_2^2) + (x_1^2 + y_1^2)}{2t(x_1 - x_2)}$$

[2]

where $K$ is the thermal diffusivity (cm$^2$ s$^{-1}$), $t$ is the time passed from the release of thermal pulse in seconds, ($x_1$, $y_1$) and ($x_2$, $y_2$) are the relative positions (cm) of the thermocouples to the heater (considering x-axis along the xylem and y-axis in the perpendicular direction) (Fig. 2).

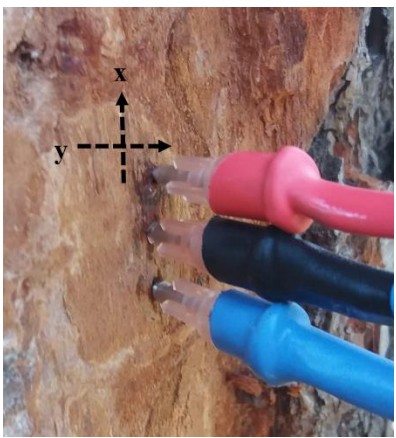

**Figure 2.** Sensor placements, from top: downstream thermocouple, heater, upstream thermocouple, in an Aleppo pine. The x-axis represents sap flow direction and the vertical distance from the heater, while the y-axis accounts for the horizontal probe distance from the x-axis. The axes should cross where the heater is inserted but are displaced for a clearer view of the coordinates.

If probes are installed equidistant to the heater and aligned along the vertical axis, then $x = x_1 = -x_2$ and $y_1 = -y_2 = 0$, equation 2 simplifies into a function that is independent of time:

$$V = \phi \ln \left(\frac{v_1}{v_2}\right) \qquad [3]$$
$$\phi = \frac{K}{x}$$

where $\phi$ is a priori, a constant only depending on the placement of the probes and on the thermal diffusivity of both the xylem and the material used in the sensors, and x, a positive magnitude representing the distance from the heater element for the two probes.

For a whole sequence of instantaneous measurements, typically being one per second from the 60[th] second to the 100[th] second after a heat pulse release, the above equation should provide a heat pulse velocity slightly dependent on time as provided by Burgess et al. (2001), when wound width and sap velocities are small, with some inherent departures being explained only by instrumental errors. The "perfect symmetry" assumption renders that the heat pulse ratio remains constant with time if heat pulse velocity (*V*), thermal diffusivity (*K*) and probe positions (in both, x and y directions) have negligible variations during the time following each heat pulse (Marshall, 1958). Burgess et al. (2001) further demonstrated how empirical results initially differed from the ideal approach described by equation 3 due to the blocking of xylem vessels and probe misplacement. However, the study concludes that the HPR converges asymptotically to a slightly tilted straight line, at least 60 seconds after the heat pulse release and, for at least 40 seconds more (until 100 seconds after the heat pulse release), which is when the instantaneous heat pulse ratio should be measured because the temperature stabilises at this point. Our study argues that a visual inspection of heat pulse velocities (*V* in equations 2 and 3) does not necessarily give enough information to decide if measured values are a good representation of the sap flow. The method does not consider that measurement errors can arise, which due to the sensitivity of the method, is likely to occur in practice. On these premises, we have built a methodology to quality check sap flow measurements systematically by introducing a statistical analysis performed on the instantaneous heat pulse ratio, acquired between 60 and 100

seconds after the heat pulse release. Hereafter, we will denote the averaged heat pulse ratio between 60 and 100 seconds as HPR. The quality check consisted of establishing threshold values for the slope of HPR against time and relative standard deviation (RSD), statistically defined as the standard deviation divided by the mean. The statistical information obtained would account for any deterioration of the measurement. Burgess et al. (2001) proposed two separate methods to correct for wounding and misalignment. The methods assume that errors arising from the wound inflicted by a sensor probe can be estimated using an empirical factor, whereas a misalignment of the probe needs to be calculated *in situ*. We propose a development of the misalignment correction method, while arguing that a statistical check of the HPR values would detect a deterioration of the signal caused by a worsening of the wound. The RSD was therefore chosen as a quality-check parameter along with the slope, which was a parameter proposed by Burgess et al. (2001).

### 3.4 Logging specifications

The proposed analysis to ensure the reliability of long time series of sap flow measurements require the storage of statistics that are not usually recorded (as it is considered unnecessary). The datalogger must have a minimum performance of the storage capacity (memory), and the processing speed of the algorithm implemented (especially in the routines related to the statistical calculations to be performed). In this study a CR800 (Campbell Scientific, USA) was used. A flow chart of sampling and data log (Fig. S1) was specifically designed, programmed and implemented in the datalogger to enable the calculations that are presented in the next sections of this paper.

We selected RSD (%) and Slope (s$^{-1}$) of the instantaneous heat pulse ratio versus time, calculated for each of the periods from 60 to 100 seconds to filter out measurement errors (Fig. 3). The HPR should be close to constant during this period, with a small slope. Deviations from an idealized slope, positive or negative, means that the HPR is not being constant with time. All HPR with relative standard deviation > 5% and a |slope – median (slope)| > 0.003 s$^{-1}$ were removed to filter out measurements with large variability in the slope, i.e. with large deviation from the ideal slope at this velocity. The 5% threshold was chosen to ensure that 95% of the dataset could be considered in the data processing. The slope median was taken from all slope values obtained during the measurement period. The magnitude of the threshold chosen for the slope was taken from a modelled output of instantaneous ratios performed by Burgess et al. (2001), were low sap flow velocities (5 cm h$^{-1}$) combined with a small wound width (0.17 cm) were shown to display a slope of 0.001. Due to our low sap flow velocities (< 8 cm h$^{-1}$) and small probe diameter (0.12 cm), we expected the slope to be close to 0.001. The specific threshold of 0.003 was decided upon inspection of the measurements and can be modified according to needle size and magnitude of the sap velocity measured. According to Burgess et al. (2001), higher values of slope (0.01) can be expected with wound width and higher velocities.

### 3.5 Correction for long-term probe misplacement

Under the assumption of "perfect symmetry", in periods when $V = 0$ cm h$^{-1}$, HPR would be equal to 1 ($v_1 = v_2$ in equation 3). However, this is not always the case with field measurements due to misalignment of the probe, damage to xylem vessels when inserting the probes, and the different thermal properties of wood and needle. Burges et al. (2001) proposed a methodology applied to *in situ* HRM measurements to account for this type of probe misalignment, building on the assumption that errors arising from inaccurate probe spacing can be treated one-dimensionally. This approach assumes that the total effect of probe misalignment (in both axes directions),

observed in the ratio of the increase in temperature, can be parameterised calculating an "effective" probe misalignment in only one direction (x direction, parallel to the xylem). Thus, without the assumption of "perfect symmetry", in periods when $V = 0$ cm h$^{-1}$, equation 2 takes a more simplified form:

$$4Kt \ln\left(\frac{v_1}{v_2}\right) = (x_2^2 + y_2^2) - (x_1^2 + y_1^2)$$

[4]

Which becomes equation (5) if $y_1 = -y_2$ or $y_1 = y_2$.

$$4Kt \ln\left(\frac{v_1}{v_2}\right) = x_2^2 - x_1^2$$

[5]

As it is unknown if the misalignment is in $x_1$ or in $x_2$, the calculation is repeated twice in the present study, assuming first that $x_1$ is correct when estimating $x_2$ placement, and vice versa. Both $x_1$ and $x_2$ may be considered slightly misaligned, but in this approximation, each of the two correction steps will assume that the other probe is correctly placed. Then the average of the obtained misalignments calculated for each of the probes will be considered as the resulting misalignment sought (Burgess et al., 2001). Each probe placement is calculated as:

$$x_2 = \sqrt{\left(4kt \ln\left(\frac{v_1}{v_2}\right) + x_1^2\right)}$$

[6]

Therefore, two different heat pulse velocities, $V_1$ and $V_2$, are derived (using equation 2 but with assumptions $y_1 = y_2$ or $y_1 = -y_2$) for the $x_1$ and $x_2$ obtained; and the final $V$ provided as their average.

Zero flow conditions can either be imposed artificially by severing the root or stem (Burgess et al., 2001), or found when there is no biophysical driving force (Forster, 2007). These conditions are accomplished when atmospheric vapour pressure deficit (VPD) is close to zero, soil is saturated usually after substantial precipitation, and measurements are taken at predawn (Forster, 2017). Zero flow conditions were assumed at night (22:00-03:00h Coordinated Universal Time, UTC) during days when REW > 0.75, and vapor pressure deficit (VPD) was close to zero (Fig. S2). Multiple readings were used to produce an average of each event. These conditions were limited to five occasions during the study period. Saturated soil is a necessary criterion due to the possibility of reverse flow at night-time (Forster, 2014). If not considered, low HPR values representing reverse flows can be interpreted as zero flow. If several calculations of $x_1$ and $x_2$ (equation 6), are performed during long monitoring periods, the non-intrusive approach of zero flow allows the parameterisation of misalignment as a function of time (equation 7). By applying equation 6 at varies times throughout the measurement period, it is possible to calculate a linear regression using the estimation of Burgess et al. (2001) for the obtained misalignments for each of the probes:

$$\begin{aligned} x_1 &= m_1\, d + n_1 \\ -x_2 &= m_2\, d + n_2 \end{aligned}$$

[7]

Where $d$ is the time along the measuring campaign in days, $x_1$ and $x_2$ are the positions of the probes relative to the heater element in cm, $m_1$ and $m_2$ are the slopes and $n_1\, n_2$ are the interception coefficients of the linear regressions

for probes 1 and 2, respectively. The negative sign in front of $x_2$ in equation 7 is to clarify that this probe needs a negative root square solution.

By introducing equation 7 in equation 2 and allowing the simplification $y_1 = y_2$ or $y_1 = -y_2$, two equations of the corrected heat pulse velocities are obtained as a function of the time during the measuring field campaign:

$$V_1 = \frac{4Kt \ln\left(\frac{v_1}{v_2}\right) + (m_1 d + n_1)^2 - 0.6^2}{2t(m_1 d + n_1 + 0.6)}$$

$$V_2 = \frac{4Kt \ln\left(\frac{v_1}{v_2}\right) + 0.6^2 - (m_2 d + n_2)^2}{2t(0.6 + m_2 d + n_2)} \qquad [8]$$

In accordance to what Burgess et al. (2001) proposes, our approach averages the two values obtained from equation 8 to estimate a corrected heat pulse velocity.

## 4 Results

### 4.1 Heat pulse ratios

The HPR obtained during the measurement period displayed a clear positive shift away from the theoretical ideal where the HPR would equal to one at zero flow (Fig. 3a, b). This gives an indication of the necessity of corrections, that being due to wound inflicted by the probe, misalignment, misestimation of thermal diffusivity- or stem water content. However, the HPR data by itself does not give an indication of the quality of each measurement, nor if it deteriorates over time. Therefore, the quality of the measurements was indicated by calculating the RSD and the slope associated with each HPR (Fig. 3c, d, e and f). All HPR with RSD higher than 5% were eliminated. The data points eliminated corresponded to a 1% of the total dataset. Because the HRM is built on the theoretical assertion that the instantaneous HPR is close to linear with time and slightly tilted, specifically between 60 and 100 seconds after the release of a heat pulse, the slope of the HPR should be small, although dependent on the sap velocity and wound width (Burgess et al. 2001). Our HPR dataset displayed slope values close to what Burgess et al. (2001) proposed, but there were also measurements where the slope varied substantially, and we decided to filter out HPR with |slope – median (slope)| > 0.003 s$^{-1}$. This corresponded to 12% of the original dataset (Fig. 3e).

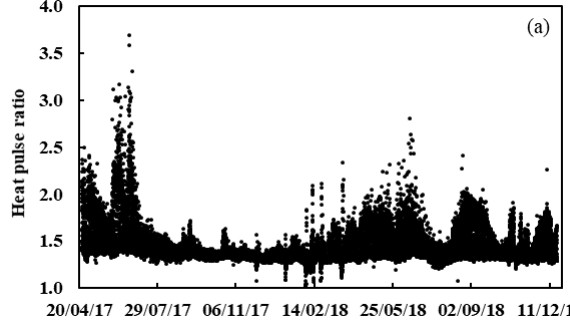 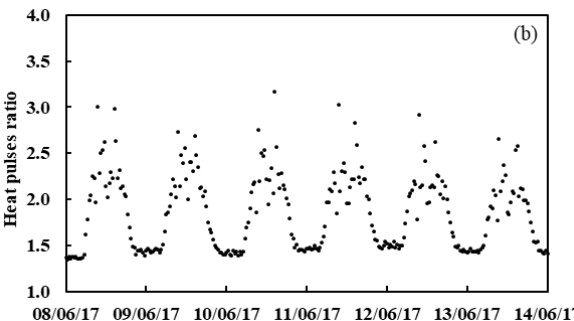

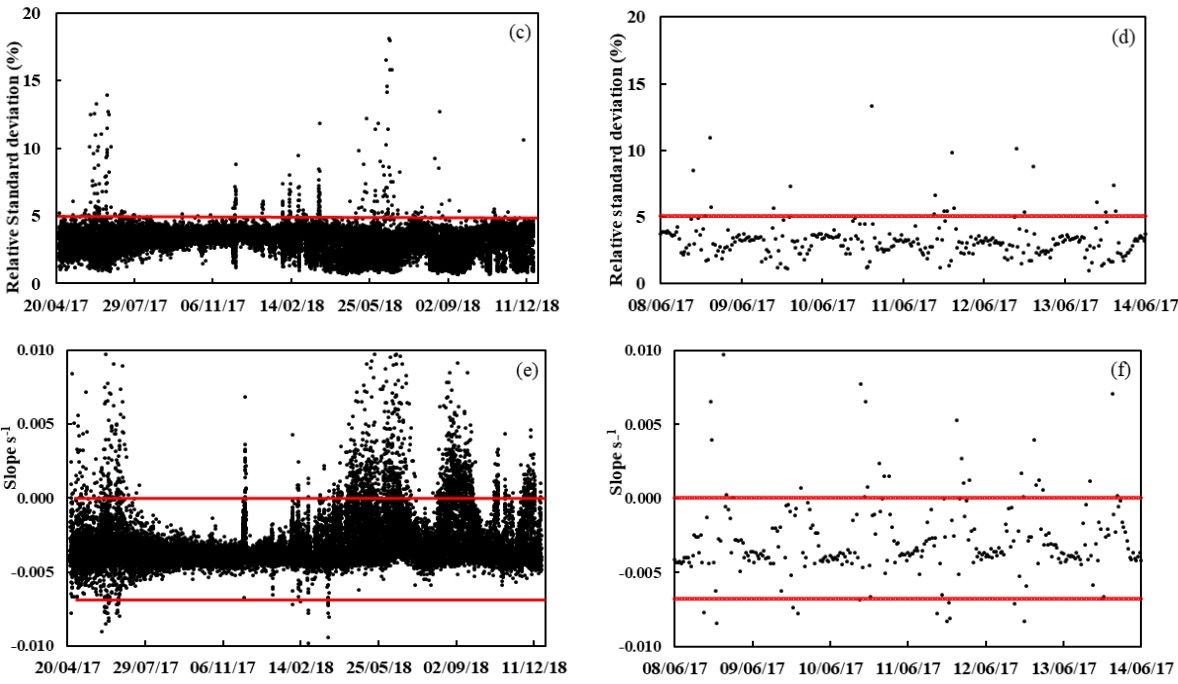

**Figure 3.** (a) Heat pulse ratios (HPR) throughout the measurement period at 30-minute intervals in pine number 1. Each HPR is an average of 41 instantaneous ratios corresponding to the temperature difference in two thermocouples at 0.6 cm up- and downstream from a heater element at 0.2 cm depth. (b) Zoomed in panel of HPR data for one week of measurements to better see the diurnal pattern. (c) Relative standard deviation (%) for each HPR in pine number 1 for the whole measurement period. Red line indicates threshold used for the quality control where all HPR relative standard deviation > 5% were removed from the data analysis. (d) Zoomed in panel of RSD data for one week. (e) Slope ($s^{-1}$) for each HPR in pine number 1 for the whole measurement period. Red line indicates threshold used for the quality control for this particular sensor. All data above or below these lines are filtered out as they did not correspond to HPR with |slope – median (slope)| < 0.003 $s^{-1}$. Median slope for this sensor was -0.004 $s^{-1}$. (f) Zoomed in panel of slope data for one week of measurements.

## 4.2 Heat pulse velocities

A linear regression was obtained from misalignment calculations for each sensor performed during zero-flow conditions. Zero-flow conditions were assumed at night (22:00-03:00h UTC) during days when REW > 0.75, and vapor pressure deficit (VPD) was close to zero (Fig. S2). Multiple readings were used to produce an average of each event. These conditions were limited to five occasions during the study period. The outputs indicate a clear shift of placement in each of the sensors over time, here denoted as $x_1$ and $x_2$ in each tree (Fig. 4). The eight probes, two per tree, all deviated from the ideal of 0.6 cm. Probe $x_2$ in pine number 1 had the highest inaccuracy with the initial value close to 0.3 cm. On average, the probes showed a shift of 0.04 cm in placement after twenty months of measurement (Fig. 4). The equation obtained from the linear regression was then implemented in equation 8, and corrections of the heat pulse velocity data was done using an average of $V_1$ and $V_2$ (Fig. 5). Outputs were compared with one-time misalignment corrections, which were calculated in the beginning of the measurement period to demonstrate the evolution of the probe misalignment. The one-time correction demonstrated a steady decline in accuracy over time (Fig. 5). The difference between the two correction methods showed significance after three months of employment (Table 1).

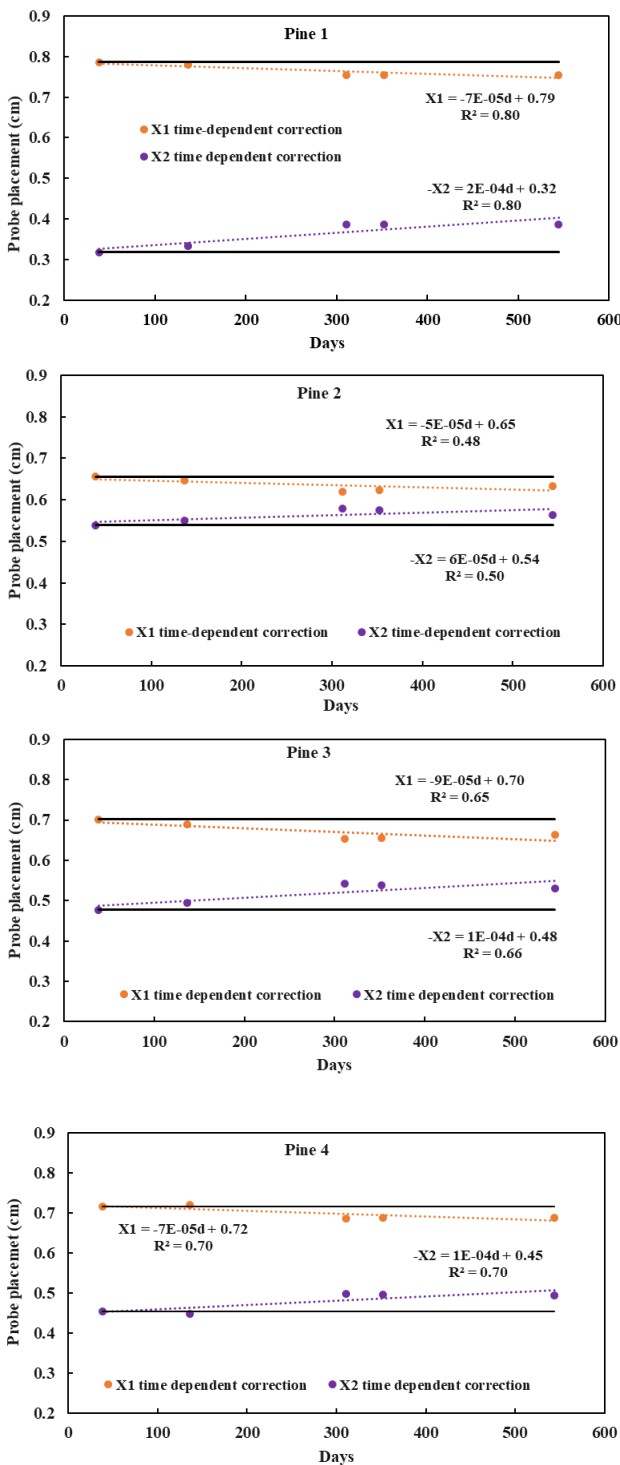

**Figure 4.** Placement of the probes are shown as distance from the heater (cm). Probe placement was calculated once at the beginning of the measurement period (solid lines) for the whole study period and compared to probe placement calculated varies times (solid circles with dotted lines) throughout the study period. Each point represents the probe position calculated during its respective zero-flow event

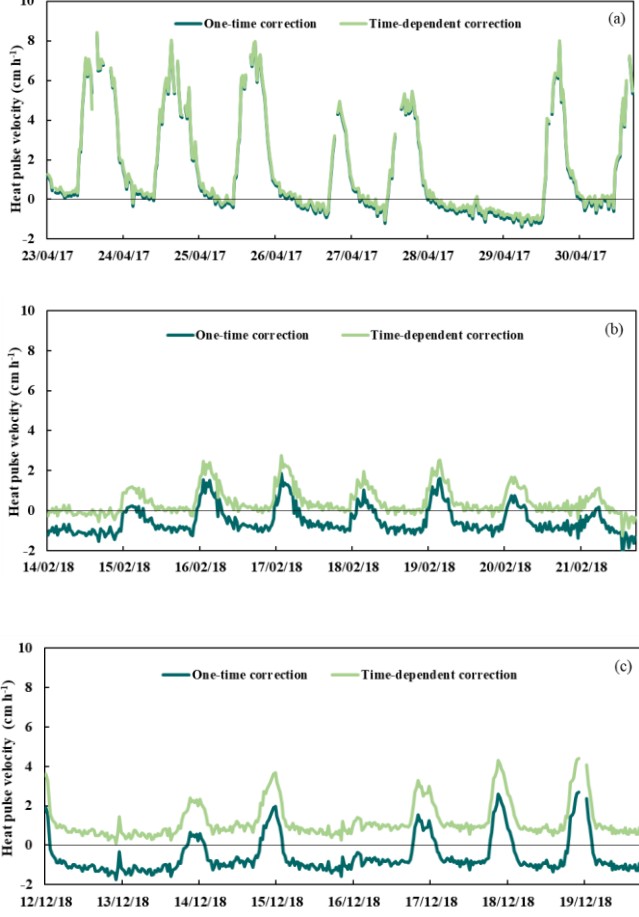

**Figure 5**. Heat pulse velocities during one week of measurements in the beginning (a), halfway (b) and at the end (c) of the measurement period. Dark green lines represent velocities with probe misalignment corrected for once at the beginning of the experiment. Light green lines represent velocities with the time- dependent probe misalignment corrections.

### 4.3 Sap flow

Heat pulse velocities (cm h$^{-1}$) were converted into sap flow (cm$^3$ cm$^{-2}$ h$^{-1}$) according to Burgess et al. (2001). Our data demonstrated that by not correcting for changes in probe misalignment under continuous measurement for more than 3 months, the errors corresponded to an averaged difference of 0.29 (cm$^3$ cm$^{-2}$ h$^{-1}$) in sap flow per quartile for the four trees. At the end of the 20-months period, this corresponded to an averaged difference of 0.53($\pm$0.23) cm$^3$ cm$^{-2}$ h$^{-1}$ (Table 1, Fig. 6). In terms of transpiration values, this corresponded to a mean difference of 0.7 L tree$^{-1}$ day$^{-1}$ (sapwood area = 170 cm$^2$), which would correspond to a difference of 542 L ha$^{-1}$ day$^{-1}$ (775 tree ha$^{-1}$) assuming 8 hours of daylight. It is relevant to note that our conversion did not consider the differences in stem moisture content, which can also affect the output values (Vandegehuchte and Steppe, 2013). The outputs obtained should be considered as relative differences as the one-time correction was applied in the beginning of the experiment.

**Table 1.** Seasonal averages of sap velocity for 4 different pines. All sap flow values are expressed in cm$^3$ cm$^{-2}$ h$^{-1}$. Sap flow corrected for with time-dependent misalignment calculations are compared with sap flow corrected for once in the beginning of the measurement period. Averages were taken from daily values. Abbreviations Sp, Su, Fa, and Wi indicates Spring (March-May), Summer (June-August), Fall (September-November) and Winter (December-February) respectively, each with the corresponding year.

| Pine | Correction method | Sp-17 | Su-17 | Fa-17 | Wi-17 | Sp-18 | Su-18 | Fa-18 |
|------|------|------|------|------|------|------|------|------|
| | One-time correction | 0.98±1.2 | 0.23±0.8 | -0.31±0.2 | -0.37±0.5 | 0.12±0.7 | 0.05±0.5 | -0.16±0.5 |

| | | | | | | | | |
|---|---|---|---|---|---|---|---|---|
| 1 | Time-dependent correction | 1.08±1.2 | 0.44±0.7 | 0.04±0.2 | 0.10±0.5 | 0.64±0.7 | 0.70±0.6 | 0.69±0.5 |
| 2 | One-time correction | 1.33±2.1 | 0.49±1.2 | -0.28±0.4 | -0.25±0.5 | 0.06±0.6 | 0.04±0.5 | -0.05±1.8 |
| | Time-dependent correction | 1.40±2.1 | 0.66±1.2 | -0.02±0.4 | 0.09±0.5 | 0.36±0.7 | 0.42±0.5 | 0.45±2.0 |
| 3 | One-time correction | 0.82±1.7 | 0.73±1.1 | -0.02±0.5 | -0.24±0.5 | 0.57±0.6 | 0.75±0.5 | 0.53±0.60 |
| | Time-dependent correction | 0.84±1.7 | 0.79±1.1 | 0.09±0.5 | -0.09±0.5 | 0.76±0.6 | 0.99±0.5 | 0.80±0.63 |
| 4 | One-time correction | 0.79±1.1 | 0.56±0.7 | 0.09±0.4 | -0.05±0.4 | 0.29±0.6 | 0.18±0.5 | 0.05±0.5 |
| | Time-dependent correction | 0.82±1.1 | 0.66±0.7 | 0.29±0.4 | 0.23±0.4 | 0.61±0.6 | 0.56±0.5 | 0.55±0.5 |

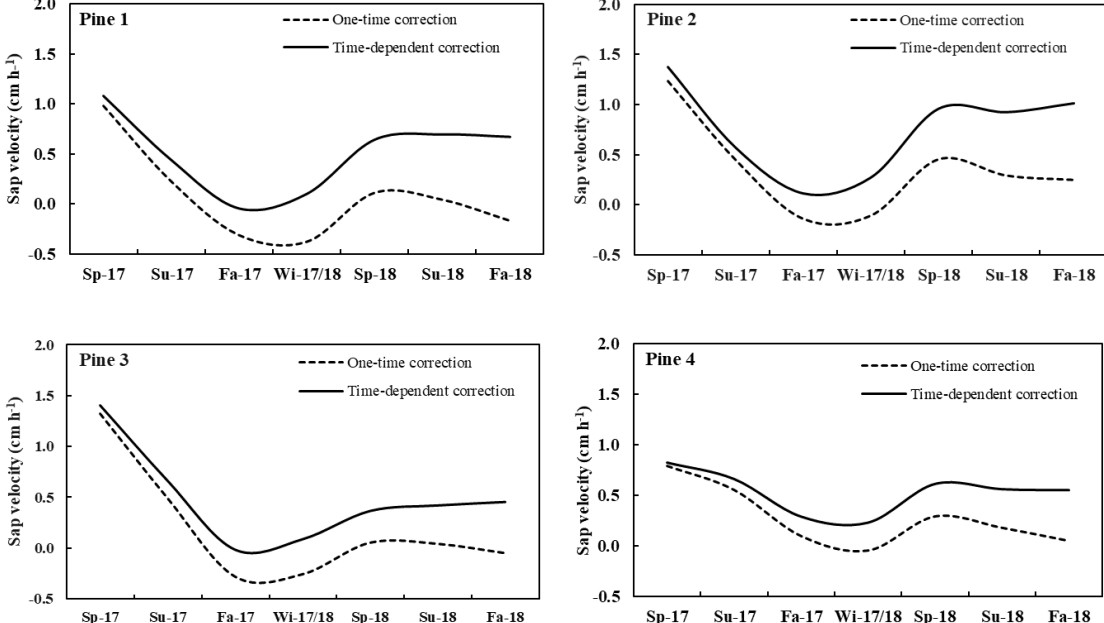

**Figure 6.** Seasonal averages of sap flow ($cm^3$ $cm^{-2}$ $h^{-1}$) calculated using the two misalignment corrections throughout the 20-month measurement period. Dotted lines represent probe misalignment corrected for once. Solid lines represent sap flow corrected for with the time-dependent probe misalignment method. Abbreviations Sp, Su, Fa and Wi indicates Spring (March-May), Summer (June-August), Fall (September-November) and Winter (December-February) respectively, each with the corresponding year.

## 4.4 Transpiration

To demonstrate the difference in terms of transpiration, sap flow values ($cm^3$ $cm^{-2}$ $h^{-1}$) without misalignment correction (T), with one-time correction ($T_{one}$), and with time-dependent correction ($T_{time}$) were converted into transpiration values (L $tree^{-1}$ $h^{-1}$) and compared against each other during a period of three weeks. Data from three of the pines (1, 3 and 4) were taken towards the end of the measurement period, to demonstrate the greatest differences (Fig. 7). However, the sensor placement in pine number two demonstrated a misplacement closer to target (0.6 cm) towards the end of the measurement period (Fig. 4), and another measurement period was therefore chosen for this pine, for a better demonstration of the differences between the estimations. Note that the transpiration estimations without misalignment correction still went through the wound correction step as shown in Burgess et al. (2001). Pine number one displayed the biggest differences between the T and $T_{time}$, with an average daily difference of 1.5±0.002 (L $tree^{-1}$) during the three weeks, and a difference in daily average of

0.3±0.003 (L tree$^{-1}$) between $T_{one}$ and the $T_{time}$. Pine number 3 and 4 showed similar differences between the methods, with a daily average of 0.1±0.001 and 0.2±0.002 (L tree$^{-1}$) respectively, between T and $T_{time}$. Both had a difference in daily average of 0.1±0.001 (L tree$^{-1}$) between $T_{one}$ and $T_{time}$ (Fig. 7).

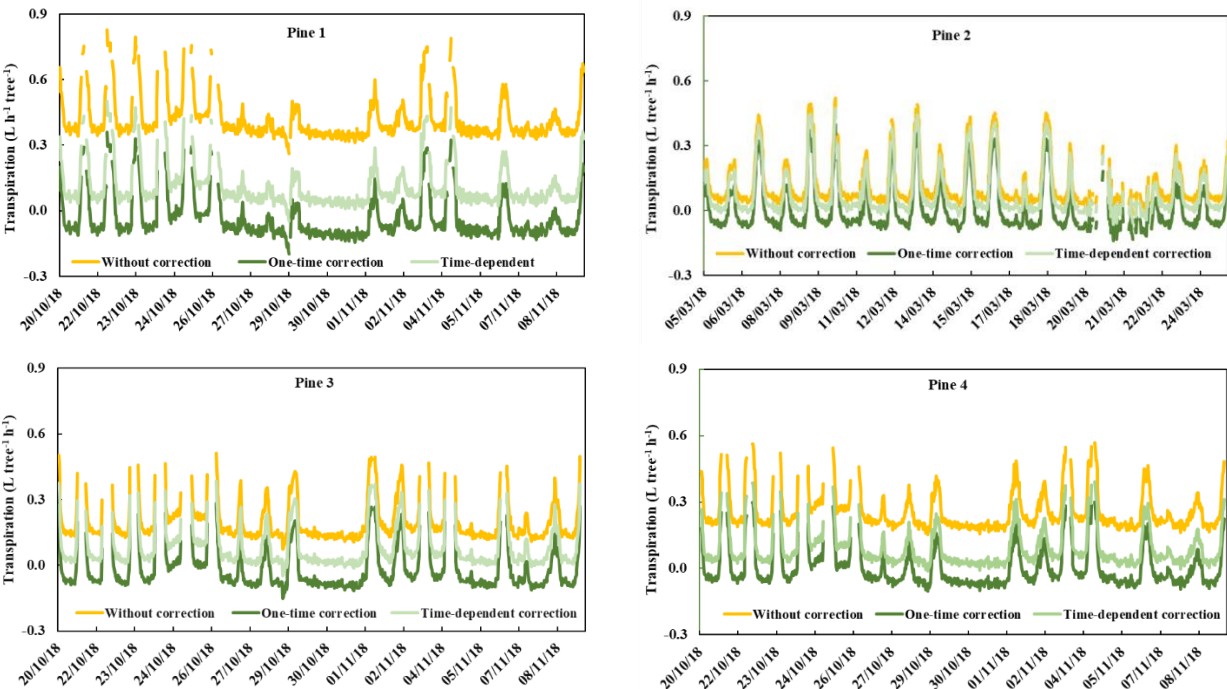

Figure 7. Three weeks of transpiration rates (L tree$^{-1}$ h$^{-1}$). 'Without correction' represents rates without misalignment-corrections, 'One-time correction' represents rates corrected for once in the beginning of the measurement period, and 'Time-dependent correction' represents rates being corrected for at varies days throughout the measurement period.

## 5    Discussion

### 5.1 Filtering heat pulse ratios

Because the HPR is an average of instantaneous temperature difference ratios, it is difficult to ensure its accuracy only by visual inspection of the averaged output, and without further statistical information. For each heat pulse measurement sequence, we suggest RSD and slope from linear regression of the instantaneous ratios versus time to filter out erratic measurements, and to ensure the quality of the data. When this procedure is considered for long-time field measurements, a clear tendency may be observed of higher RSD and slope values during periods of higher flow rates (Fig. 3), indicating more noise and less trustworthy data during those episodes. The HRM only measures values below 45 cm h$^{-1}$ (Forster, 2017), due to a maximum ratio of $\frac{v_1}{v_2} = 20$, for when the ratio can be assumed accurate (Burgess et al., 2001; Marshall, 1958). Because our dataset showed no higher velocities than 8 cm h$^{-1}$, it was not initially considered a limitation for the measurements.

During twenty months of field measurements, the thermocouples showed no visible sign of deterioration with time. However, it is still important to note that this does not consider the possible diminishing amplitude of the ratios over time, which can also lead to underestimations of actual flow (Barett et al., 1995; Green et al., 2003; Forster, 2017). Therefore, this should be observed separately, comparing sap flow ratios using data obtained under similar climatic conditions (Moore et al., 2010). Due to the variation in rainfall between the two years, SWC differed significantly on similar calendar dates apart from six days in June (9 – 15). When comparing the

relationship of sap flow versus VPD on these days, there was an increase in the slope; 2.3 for 2017 and 2.9 in 2018. This, we attributed to an overall weaker correlation between VPD and sap flow in 2017; $R^2 = 0.4$ in 2017 versus $R^2 = 0.6$ in 2018, due to higher values of VPD the first year (Fig. S3). In conclusion, it was clear that the HPR readings had not decreased in the second year, when compared to the first year, under similar environmental conditions.

**5.2 Long term variation in probe placement**

When applying the HRM for longer than a few weeks it is relevant to quantify how $\boldsymbol{\phi}$ in equation (3) and the misalignment term in equation (5) evolve during the measuring period. The predicted variation of $x_1$ and $x_2$ is due to the growth of the tree and, on the other hand, periodical variations in $K$ due to annual and seasonal variations of the physiological properties of the tree (Green et al., 2003; Vandegehuchte and Steppe, 2012; Ren et al., 2017). In the HRM, the probe misalignment calculations can be corrected using the methodology proposed here, considering that as an average, each sensor displayed a 0.04 cm displacement within the tree after twenty months of measurements. The correction would also be more rigorous with added zero flow events, which would be easier to obtain in humid environments with more periods of saturated soil.

The theory behind the HPV method were tested on conifers (Marshall, 1958), whereas Burgess et al. (2001) more generally refers to woody species when working with the HRM. The time-dependent correction method could be useful with any species already tested with the HRM, where misalignment of probes can create a source of error and sensors are installed over a longer period. Specifically, where the movement of the wood might cause further displacement of the sensor.

By going back to the original assumption of "perfect symmetry", we investigated the original premises the method is built on. Even though Burgess et al. (2001) elaborated a correction method for sensor misalignment we saw that changes in sensor placement were detectable after each season. Therefore, multiple corrections should be carried out throughout the measurement period. The proposed modified method coincides with the one-time correction method (Burgess et al., 2001) for short time periods (< 3 months) but differs progressively over time. We found that this shift in placement is significant already after 3 months, and therefore dynamically misplacement calculations should be carried out or sensors should be reinstalled at this frequency. However, as pointed out by Moore et al. (2010), re-installing the sensors might create a shift in the data due to spatial variation within the tree. Leaving the same sensors in the tree throughout the study period avoids this problem and enables the study to focus on the intrinsic factors affecting the sap flow rates.

**2. Conclusion**

In conclusion, we found that high quality measurements with sap flow sensors can be ensured over longer periods (>3 months), if the HPR is assessed using the proposed filtering method, and the probe misalignment variability over time corrected for. In this study, we observed data over a 20-month period in *Pinus halepensis*, and saw no sign of deterioration in the second year compared to the first, when observing the values obtained for slope and relative standard deviation. However, when observing the alignment of each probe, there was a clear shift from the beginning to the end of the measurement period. This indicate that measurements can be obtained during a

second season without the need of re-installing sap flow sensors, if the proposed time-dependent misalignment correction is incorporated in the data processing. This would increase the accuracy of point measurements, and consequently transpiration estimations. The different errors related to upscaling are beyond the scope of this paper, but significant differences were observed when comparing sap flow estimations with no correction, one-time correction, and time-dependent correction for probe misalignment.

## Author contributions

JLP and ECH conceptualized the study. EKL and JLP prepared the manuscript with inputs from all the co-authors. EKL made the sensors with the supervision of JB. EKL implemented the sensors and had the responsibility of the data processing and field site. JLP, EKL and JAV all worked on the data analysis. JAV wrote the script for the datalogger and had the technical responsibility. JLP and JAV designed the flow chart to be programmed in any data logger system.

The authors declare that they have no conflict of interest

## Acknowledgements

Fundacion CEAM is supported by the Generalitat Valenciana (Spain). This study is partially funded by the Spanish Ministry of Science, Innovation and Universities and the European Regional Development Fund CGL2015-67466-R (MICINN/FEDER) through the research project "VERSUS", by the project "IMAGINA" PROMETEU/2019/110 (Consellería de Cultura GVA), and by the research project "DESESTRES"–PROMETEOII/2014/038 (Consellería de Cultura GVA). We thank the municipality of Aras de los Olmos for permitting field experiments on a public managed forest site. We thank J. Puértolas and H. Moutahir for their feedback and encouragement on the manuscript.

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
