# Peer review of "Technical note on long-term probe misalignment and proposed quality control using the heat pulse method for transpiration estimations"

_Hydrology and Earth System Sciences, 2019_

## Referee Comment (RC1) · Anonymous Referee #1 · 17 Jul 2019

General comments:

The manuscript entitled, Technical note on long-term probe misalignment and proposed quality control using the heat pulse method for transpiration estimations, addresses an important technical issue with regards to probe misalignment. It is suitable as a technical note within HESS and it delivers a concrete way of dealing with probe misalignment. More specifically, it provides a concrete statistical filtration method to ensure the consistency of measurements. The work provides a substantial contribution to scientific progress within the sap flow community. The methods are valid, yet more clarification is needed with regards to the sampling design and a more elaborate

analyses are needed on the individual specific effects. Although the results and con-
clusions are presented clearly, there is a need within the manuscript to elaborate on
the implications of the results and potential other issues present with the methods that
is applied.

The main issue I see with regards to the manuscript is that the authors could spend
more attention on the variability of misalignment issues between installed trees and
present in a clear way how they defines zero flow conditions. Additionally, the implica-
tions of the correction could be more concretely quantified by showing time-series of
daily water use, with and without the proposed correction. Finally, within the discussion
there is space for further elaborating on the other issues related to the installment of
these type of sensors. Clearly circumferential variability, wounding and other biases
should be further investigated in the future.

Specific comments:

Line 44-46: Please provide sources describing each of the method. Now readers
cannot read related literature to explain the method.

Line 56-59: See also: Steppe et al. 2010 A comparison of sap flux density using
thermal dissipation, heat pulse velocity. This is a relevant study which addresses the
offset of sap flow methods from gravimetric measurements.

Figure 1: It would be good if there would be a zoom in panel where you can see more
detail on the patterns. The current point cloud does not give the reader a full idea
on the diurnal quality of the data. Line 220: Please help the reader to understand
what RSD is. I went back to the methods to check, yet this part of the text should be
understandable on its own.

Line 237-238: It would be good to understand when and why there is a limited amount
of periods with zero-flow conditions. Additionally, it is not clear how these periods where
exactly defined. It would help to include an appendix figures which details these periods

and the underlying environmental conditions. Additionally, now the displacement is provided for one tree and the average of all trees and sensors. Yet, it would be good to see whether there are difference between the sensors themselves. Would the authors be able to provide the change presented in Figure 2 for each sensor?

Line 263-265: It would be interesting to see temporally what the difference are in daily water use (L d-1). This will clarify if the offset due to misalignment is progressively getting worse or whether, in these species, the impact is not that bad. Additionally, when presenting these numbers it is critical that the standard deviation is also provided for these estimates.

Line 296-299: Indeed, there could be a reduction in the amplitude due to wounding effects or other changes within the stem. Did the authors analyse whether they would see a reduction in the amplitude over time? It would be important to make this test as the data is available.

Line 316-318: This is indeed a valid point, yet I would propose that the authors would elaborate on the fact that reinstalling sensors along the stem will introduce change due to circumferential variability in the stem. This could be critical when generating continuous series of sap flow over the long term. Also, do the authors think these results found on conifers are universal for all types of species? I would have expected a short discussion to clarify to the reader why these findings could be of general value to the application of the method.
* * *

---

## Referee Comment (RC2) · Christoforos Pappas (Referee) · 14 Aug 2019

Overview: Larsen et al. in this Technical note address two important issues of sap flow measurements with the heat pulse method, namely (i) data filtering/quality control of the raw heat pulse records, and (ii) errors due to misalignment of the sap flow probes. The authors suggest some statistical thresholds/filters to be applied in the raw heat pulse ratios for data cleaning and present a time-dependent correction to account for probe misalignment. Moreover, they demonstrate the importance of such uncertainties for robust transpiration estimates, by presenting sap velocity and transpiration estimates with and without applying the proposed correction. I find the study topical

and of interest for the scientific community working on transpiration estimates with the heat pulse sap flow method. However, I feel that the manuscript needs major revisions to better present the motivation and rationale of the study, the developed methods, and the broader implications of the obtained results.

Main comments:

1. The text needs significant editorial improvements to eliminate vague/unclear wording and grammatical errors. Moreover, several parts of the text (including the abstract) need to be revised/rephrased/rewritten to improve the clarity of the text and better communicate the design of the study, results, discussion and conclusions. I have highlighted few specific points below (see Specific comments), yet several other cases exist throughout the manuscript.

2. The methods need to be revised and clarified. In some parts, there are inconsistencies and it is hard to follow. Sometimes the authors refer to V as sap velocity (L130) and other times as heat pulse velocity (L113, L135). Please clarify and use consistently the terms/variables/abbreviations throughout the manuscript. Also, the selected thresholds (L153-161) for the raw data filtering need to be better justified, since at the moment seem quite arbitrary or could be interpreted as case-specific. Also, the data from all eight sensors (or averages across trees, since two sensors per tree were deployed) should be presented, either in the main text, or in the supplementary material. Apart from Fig 5, all figures illustrate data from a single sensor. In addition, more details should be provided in the methods session on how the positions of sensor misalignment were estimated in Fig 2 (and the misalignment for all eight sensors would be interesting to be illustrated, too).

3. I feel that the hydrological community and the readership of this Journal, would appreciate also some figures with the up-scaled transpiration estimates and the resulting biases do to probe misalignment, complementing the existing figures with the sap velocities and the results presented in L265-268.

4. The suggested time-depended correction accounts for two effects: probe misalignment and wounding effects. The current experimental design does not allow to disentangle the two. Therefore, the text should be revised so it is clear that the proposed correction addresses issues related to both wounding and probe misalignment.

5. I suggest to include a comparison between sap velocities/transpiration estimates averaged throughout the study period/growing season as calculated with (i) no wounding correction, (ii) traditional (no time depended) corrections, and (iii) the presented time-depended correction. This would better emphasise/illustrate the advantages of this Technical note.

Specific comments:

Abstract: the study location, tree species, number of instrumented trees, study period should be clearly stated in the abstract.

L16: 'Whole-plant transpiration' reads redundant, just 'Transpiration' should be enough here.

L17: and Hydrology.

L18: 'wide application range' and L19: 'ready automation': unclear what you mean here. Please consider revising/rephrasing. Similar for 'data readings', I guess what you mean here is the sap flow sensors can provide long-term measurements of sap flow in tree stems with high temporal resolution (e.g., minutes, hours etc.).

L19: 'Several different': reads redundant. 'Several methods' or 'Different methods' should be enough. L20: how the methods were adjusted to different climatic conditions? Unclear statement. Maybe 'tested' instead of 'adjusted'?

L21: 'in the method', unclear to which method you are refereeing to, here. Please rephrase/revise.

L21-22: if you focus only on the heat pulse method, then that is probably fine, but if

you are referring to sap flow methods in general, then additional sources of uncertainty should be listed here, e.g., Granier's empirical parameters, zero-flow conditions, see for example:

Flo et al. (2019), A synthesis of bias and uncertainty in sap flow methods, AFM, 10.1016/j.agrformet.2019.03.012, 271.

Peters et al. (2018), Quantification of uncertainties in conifer sap flow measured with the thermal dissipation method, New Phytol., doi:10.1111/nph.15241.

L23: 'readings' is not the right word here. The readings are what is recorded in the data logger, the proposed method is a data-preprocessing method that can improve the final sap flow estimates, and ultimately the whole-plant transpiration values.

L25: 'imply' is not the right wording here.

L26: 'statistical record to be recorded' unclear what you mean here, please revise/rephrase.

L27-29: This sentence is hard to follow: standard deviation and slope of which quantity? Please revise/clarify.

L40-42: to which direct measurements of transpiration you are referring to here? Please clarify.

L50-54: I cannot follow how the eddy-variance method comes to the discussion here. Eddy-covariance measurements of latent heat are not tree transpiration measurements, but evapotranspiration measurements at the landscape level (including soil evaporation, evaporation from interception, transpiration from over- and under-story, etc.).

L90-100: mentioned that you deployed eight sensors in total, two per tree. I found this information further below in the text, but this has to be very clear from the methods session.

L94-97: mention the specific depth where the thermocouples are located, and thus the heat velocity is measured. I found this information mentioned in a figure caption (L229) but has to be included in the methods description.

L138: you are referring to the raw heat velocities here I assume and not to sap flow measurements. Here and throughout the text clarify and use carefully and consistently terms such as heat velocity, sap velocity, and sap flux density.

L154: 'ratio' of what?

L160-161: please provide more details on how the specific threshold for the slope was selected/defined.

L286 'Filtration' please change to 'filtering' or 'pre-processing'

---

## Author Comment (AC1) · 16 Sep 2019

We would like to thank the referee for concise and constructive feedback. We will address these issues in the revised manuscript, as according to our responses to each comment below. Our response will be highlighted in blue below the referee's comments in black.

The main issue I see with regards to the manuscript is that the authors could spend more attention on the variability of misalignment issues between installed trees and present in a clear way how they define zero flow conditions. Additionally, the implications of the correction could be more concretely quantified by showing time-series of daily water use, with and without the proposed correction. Finally, within the discussion there is space for further elaborating on the other issues related to the installment of these type of sensors. Clearly circumferential variability, wounding and other biases should be further investigated in the future.

We have gone through and corrected for the issues the referee points out. Because this is a technical note and we didn't want to increase the figures-to-text ratio too much, some of the additional figures are added in the supplementary materials. Figures are here shown below the text.

Line 44-46: Please provide sources describing each of the method. Now readers cannot read related literature to explain the method.

We've included references in the text for each method mentioned in the text:

> "There are a range of different approaches to sap flow measurements, and methods vary between
>
> heat dissipation (HD), (Granier, 1987; Lu et al., 2004), steam heat balance (SHB), trunk segment
>
> heat balance (THB), (Smith and Allen, 1996), and heat pulse velocity (HPV), (Marshall, 1958).
>
> However, they are all based on tracing heat within the xylem (Burgess et al., 2001; Davis et al.,
>
> 2012; Forster, 2017)."

Line 56-59: See also: Steppe et al. 2010 A comparison of sap flux density using thermal dissipation, heat pulse velocity. This is a relevant study which addresses the offset of sap flow methods from gravimetric measurements.

Thank you for providing the reference, even if the paper is comparing the overall heat pulse velocity technique rather than the specific heat ratio method, we found it very relevant and decided to include the reference:

> "By accounting for these sources of error and additionally estimating the stem moisture content and
>
> radial variability, the heat ratio method (HRM) has been evaluated the heat pulse velocity method
>
> (HPV) with highest accuracy, although with a tendency of underestimating transpiration values
>
> (Forster, 2017), an error that is shown to increase with higher sap flow values (Steppe et al., 2010)."

The paper also made us aware of an error of concept; using the term "sap velocity" and the unit cm h$^{-1}$ is not precise, and we have therefore changed it throughout the paper to "sap flow" per unit of sap wood (cm$^3$ cm$^{-2}$ h$^{-1}$).

Figure 1: It would be good if there would be a zoom in panel where you can see more detail on the patterns. The current point cloud does not give the reader a full idea on the diurnal quality of the data.

As suggested by the referee, we have included a zoomed in panel in Fig.1, representing one week of data for the HRP, slope and RSD. We agree that it enhance the understanding of the filtration as it gives the reader the diurnal pattern of the data and therefore demonstrates the range of values that has been filtered out.

Line 220: Please help the reader to understand what RSD is. I went back to the methods to check, yet this part of the text should be understandable on its own.

We included a definition for the RSD also in this part of the text so the reader doesn't need to go back in the text to check, specifically:

> "Therefore, the quality of the measurements was indicated by calculating the RSD, the relative
>
> standard deviation divided by the sample mean, and the slope versus time, for each HPR."

Line 237-238: It would be good to understand when and why there is a limited amount of periods with zero-flow conditions. Additionally, it is not clear how these period where exactly defined. It would help to include an appendix figures which details these periods and the underlying environmental conditions.

As suggested by the referee, a definition of zero-flow conditions was included. The definition now references five additional figures in the appendix, showing the relative extractable water and VPD along with the heat pulse ratios during the estimated zero-flow events:

"Zero-flow conditions were assumed at night (22:00-03:00h solar time) during days when relative extractable water, REW > 0.75, and vapor pressure deficit (VPD) was close to zero (Fig. S2). Multiple readings were used to produce an average of each event. These conditions were limited to five occasions during our study period."

An additional paragraph was included in the method section due to the inclusion of environmental data:

"Environmental conditions

Air relative humidity (%) and air temperature (ºC) were registered every 30-minute (U23 Pro V2, Onset Computer Corporation, USA). Precipitation was registered using a rain gauge with 0.2 mm resolution (RG3-M, Onset Compute, USA). Three soil moisture probes were inserted at 20-25 cm depth to register soil water content, SWC, (S-SND.M005, Onset Computer Corporation, USA), using a datalogger for logging specifications (HOBO Micro Station, USA). To assume periods of zero-flow events, relative extractable water (REW) was calculated using the method of Bréda et al. (1995):

$$REW = \frac{(\theta_t - \theta_{min})}{(\theta_{max-} \theta_{min})} \qquad [1]$$

Were $\theta_t$ is the registered SWC, $\theta_{min}$ is the minimum SWC observed during the measurement period, and $\theta_{max}$ is the SWC at field capacity. When values of REW surpassed 1 they were converted to 1 according to Granier et al. (2000)."

Additionally, now the displacement is provided for one tree and the average of all trees and sensors. Yet, it would be good to see whether there are differences between the sensors themselves. Would the authors be able to provide the change presented in Figure 2 for each sensor?

On request from the referee, we decided to include the change described in Figure 2 for each sensor. We also included a better definition of the sensor to clarify that each sensor consists of three needles, two probes and one heater. When referring to misalignment we refer to both probes in a sensor.
 In the text:

"Each sap flow sensor consists of three needles: one heater and two thermocouples. We will refer to the thermocouples as probes, and when using the term "sensors" we refer to both probes and the heater."

And regarding the misalignment:

"The outputs indicate a clear shift of placement in each of the probes over time, here denoted as $x_1$ and $x_2$ in each pine (Fig. 2). The eight probes, two per tree, all deviated from the ideal of 0.6 cm. Probe $x_2$ in pine number 1 had the highest inaccuracy with an initial value close to 0.3 cm".

Line 263-265: It would be interesting to see temporally what the difference are in daily water use (L d-1). This will clarify if the offset due to misalignment is progressively getting worse or whether, in these species, the impact is not that bad. Additionally, when presenting these numbers, it is critical that the standard deviation is also provided for these estimates.

The authors originally didn't include the difference of daily water use (L d$^{-1}$) because this introduces the errors related to upscaling to whole tree transpiration, whereas we wanted to focus on the correction of the point measurements. However, we chose to include three weeks of data towards the end of the measurement period, also including the transpiration values without any misalignment correction (Fig. 6). This also highlighted a very small difference between the "non-correction" method and "time-dependent-correction" when the misalignment was small or converged towards the ideal distance of 0.6 cm between the probes. In the original manuscript, Fig. 5 represents how the difference between the two methods changes in time and is meant to give an indication of how the misalignment estimation is getting progressively worse if the misalignment is measured only once at the beginning of a measurement campaign. Line 266-268 in the original manuscript shows the overall difference between using the correction during the entire study period of 20 months. Standard deviation was provided for the estimates in table 1.

Line 296-299: Indeed, there could be a reduction in the amplitude due to wounding effects or other changes within the stem. Did the authors analyse whether they would see a reduction in the amplitude over time? It would be important to make this test as the data is available.

We have taken the suggestion of the referee into consideration and included the test results in the paper and the figures in the supplementary material:

> "Comparing the data from 2017 to the data of 2018, SWC differed significantly on similar calendar dates with the exception of six days in June (9 – 15). When comparing the relationship of sap flow versus VPD on these days, there was an increase in the slope the second year; 2.3 for 2017 and 2.9 for 2018. This, we attributed to an overall weaker correlation between VPD and sap flow in 2017; $R^2$ = 0.4* in 2017 versus $R^2$ = 0.6* in 2018, due to higher values of VPD in 2017 (S3). Within the normal variability of the measured data we concluded that the sap flow values had not decreased in the second year, when compared to the first year, under similar environmental conditions."

Line 316-318: This is indeed a valid point, yet I would propose that the authors would elaborate on the fact that reinstalling sensors along the stem will introduce change due to circumferential variability in the stem. This could be critical when generating continuous series of sap flow over the long term.

We appreciate the referee pointing this out as it is an important point to make:

> "However, as pointed out by Moore et al. (2010), re-installing the sensors might create a shift in the data due to spatial variation within the tree. Leaving the same probes in the tree throughout the measurement period avoids this problem and enables the study to focus on the intrinsic factors affecting the sap flow pattern."

Also, do the authors think these results found on conifers are universal for all types of species? I would have expected a short discussion to clarify to the reader why these findings could be of general value to the application of the method.

We agree that it's an important addition to the discussion and we have therefore included it in the text:

> "The time-dependent correction method could be useful for any species already tested with the HPR, where misalignment of probes can create a source of error and sensors are installed over a longer

period of time. Specifically, where the movement of the wood might cause further displacement of the sensor."

Corresponding figures below:

[Figure]

[Figure]

[Figure]

Figure 1. (A) Heat pulse ratios (HPR) throughout the measurement period in 30-minute intervals in pine number 1. Each HPR is an average of 41 instantaneous ratios corresponding to the temperature difference in two thermocouples at 0.6 cm up-and downstream from a heater probe at 0.2 cm depth. (A1) Zoomed in panel of HPR data for one week of measurements. (B) Relative standard deviation (%) for each HPR in pine number 1 for the whole measurement period. Red line indicates threshold used for the quality control were all HPR with relative standard deviation > 5% were removed from the data analysis. (B1) Zoomed in panel of RSD data for one week. (C) Slope ($s^{-1}$) for each HPR in tree number 1 for the whole measurement period. Red line indicates threshold used for the quality control for this particular sensor. HPR with |slope – median (slope)| < 0.003 $s^{-1}$ were removed from the data analysis. (C1) Zoomed in panel of slope data for one week of measurements.

[Figure]

[Figure]

Figure 2. The placements of the probes are shown as distance from the heater (cm). Probe placement was calculated once (solid lines) for the whole study period and compared to probe placement calculated varies times (solid circles with dotted lines) throughout the study period. Each point represents the probe position calculated during its respective zero-flow event.

Table 1. Seasonal averages of sap velocity for 4 different pines. All sap flow values are expressed in cm$^3$ cm$^2$ h$^{-1}$. Sap flow corrected for with time-dependent misalignment calculations are compared with sap flow corrected for once in the beginning of the measurement period. Averages were taken from daily values. Abbreviations Sp, Su, Fa, and Wi indicates Spring, Summer, Fall and Winter respectively, each with the corresponding year.

| Pine | Correction method | Sp-17 | Su-17 | Fa-17 | Wi-17 | Sp-18 | Su-18 | Fa-18 |
|------|-------------------|-------|-------|-------|-------|-------|-------|-------|
|   | One-time correction | 0.98±1.2 | 0.23±0.8 | -0.31±0.2 | -0.37±0.5 | 0.12±0.7 | 0.05±0.5 | -0.16±0.5 |
| 1 | Time-dependent correction | 1.08±1.2 | 0.44±0.7 | 0.04±0.2 | 0.10±0.5 | 0.64±0.7 | 0.70±0.6 | 0.69±0.5 |
|   | One-time correction | 1.33±2.1 | 0.49±1.2 | -0.28±0.4 | -0.25±0.5 | 0.06±0.6 | 0.04±0.5 | -0.05±1.8 |
| 2 | Time-dependent correction | 1.40±2.1 | 0.66±1.2 | -0.02±0.4 | 0.09±0.5 | 0.36±0.7 | 0.42±0.5 | 0.45±2.0 |
|   | One-time correction | 0.82±1.7 | 0.73±1.1 | -0.02±0.5 | -0.24±0.5 | 0.57±0.6 | 0.75±0.5 | 0.53±0.60 |
| 3 | Time-dependent correction | 0.84±1.7 | 0.79±1.1 | 0.09±0.5 | -0.09±0.5 | 0.76±0.6 | 0.99±0.5 | 0.80±0.63 |
|   | One-time correction | 0.79±1.1 | 0.56±0.7 | 0.09±0.4 | -0.05±0.4 | 0.29±0.6 | 0.18±0.5 | 0.05±0.5 |
| 4 | Time-dependent correction | 0.82±1.1 | 0.66±0.7 | 0.29±0.4 | 0.23±0.4 | 0.61±0.6 | 0.56±0.5 | 0.55±0.5 |

[Figure]

[Figure]

Figure 6. Timeline of transpiration estimates (L h[-1] tree [-1]) for each sensor and each tree during 3 weeks at the end of the measuring period. Three estimations are shown; without any correction for misalignment of the probes (yellow line), onetime correction for misalignment of the probes (dark green), and time-dependent correction for misalignment of the probes (light green). The timeline represents diurnal measurements taken at a 30-minute interval.

Supplementary material:

[Figure]

[Figure]

[Figure]

*Figure. S2. Vapour pressure deficit (VPD), relative extractable water (REW) and heat pulse ratio (HPR) for all trees at five different events assuming zero flow conditions. Squares indicate the HPR readings used for zero flow estimations, between 22:00 and 03:00 solar hours. Each panel includes five days of data.*

[Figure]

**Figure S3.** The observed relationship between averaged sap flow for four *Pinus halepensis* and vapour pressure deficit (VPD), between June 9 and June 15, for 2017 and 2018. Each point represents measurements every 30-minute.

---

## Author Comment (AC2) · 17 Sep 2019

We thank the referee for his thorough reading of our paper and his useful feedback. The text has been revised extensivley, both in terms of language and in form throughout the manuscript. Our response will be highlighted in blue below the referee's comments in black.

Overview: Larsen et al. in this Technical note address two important issues of sap flow measurements with the heat pulse method, namely (i) data filtering/quality control of the raw heat pulse records, and (ii) errors due to misalignment of the sap flow probes. The authors suggest some statistical thresholds/filters to be applied in the raw heat pulse ratios for data cleaning and present a time-dependent correction to account for probe misalignment. Moreover, they demonstrate the importance of such uncertainties for robust transpiration estimates, by presenting sap velocity and transpiration estimates with and without applying the proposed correction. I find the study topical and of interest for the scientific community working on transpiration estimates with the heat pulse sap flow method. However, I feel that the manuscript needs major revisions to better present the motivation and rationale of the study, the developed methods, and the broader implications of the obtained results.

To better present the motivation of the study we have made some changes in the text (L61):

"Previous studies have suggested additional solutions for probe misalignment (Ren et al., 2017), for determining thermal diffusivity (Vandegehuchte and Steppe, 2012), and correcting for heterogeneous heat capacity in wood (Becker and Edwards, 1999). However, there is no recent recommendation for how long newly deployed sap flow sensors can be employed. Some studies have shown how sensor probes inserted into the xylem can dampen the signal due to blocking or destruction of vessels (Moore et al., 2010; Wiedemann et al., 2016). One way to account for changes over time has been to reinstall sensors throughout the study period (Moore et al. 2010), however there is little information to be found on the exact interval for which this needs to happen (Vandegehuchte and Steppe, 2013), and for continuous measurements this will interrupt the dataset (Moore et al. 2010). Therefore, we aim to find a quality indication that can ensure that the readings don't deteriorate over time, or if they do, that it would be detectable. Attention should be given to check the accuracy of the heat pulse ratio itself, in which the rest of the methodology is built on. In addition, to allow for sensors to be employed over longer periods it's necessary to develop a dynamic probe misalignment correction method due to observed change in probe position over time."

1. The text needs significant editorial improvements to eliminate vague/unclear wording and grammatical errors. Moreover, several parts of the text (including the abstract) need to be revised/rephrased/rewritten to improve the clarity of the text and better communicate the design of the study, results, discussion and conclusions. I have highlighted few specific points below (see Specific comments), yet several other cases exist throughout the manuscript.

We have corrected specific comments and rewritten phrases in the abstract, methods, discussion and conclusion. We hope this has led to more clarity and improved the communiation of our study. As we cannot resite the whole text here, we have chosen to include the rewriting of the conclusion:

"In conclusion, we found that high quality measurements with sap flow sensors can be ensured over longer periods (>3 months), if the HPR is assessed using the proposed filtering method, and the probe misalignment variability over time corrected for. In this study, we observed data over a 20-

month period in *Pinus halepensis*, and saw no sign of detoriation in the second year compared to the first, when observing the values obtained for slope and relative standard deviation. However, when observing the alignment of each probe, there was a clear shift from the beginning to the end of the measurement period. This indicate that measurements can be obtained during a second season without the need of re-installing sap flow sensors, if the proposed time- dependent misalignment correction is incorporated in the data processing. This would increase the accuracy of point measurements, and consequently transpiration estimations. The different errors related to upscaling are beyond the scope of this paper, but significant differences were observed when comparing sap flow estimations with no correction, one-time correction, and time-dependent correction for probe misalignment. To avoid sensor reinstallation, this should therefore be considered."

2. The methods need to be revised and clarified. In some parts, there are inconsistencies and it is hard to follow. Sometimes the authors refer to V as sap velocity (L130) and other times as heat pulse velocity (L113, L135). Please clarify and use consistently the terms/variables/abbreviations throughout the manuscript. Also, the selected thresholds (L153-161) for the raw data filtering need to be better justified, since at the moment seem quite arbitrary or could be interpreted as case-specific. Also, the data from all eight sensors (or averages across trees, since two sensors per tree were deployed) should be presented, either in the main text, or in the supplementary material. Apart from Fig 5, all figures illustrate data from a single sensor. In addition, more details should be provided in the methods session on how the positions of sensor misalignment were estimated in Fig 2 (and the misalignment for all eight sensors would be interesting to be illustrated, too).

Inconsistencies highlighted by the referee has been addressed and corrected for. Further inconsistencies or vague formulations has been checked throughout the manuscript. We argue that the selected threshold needs to be case-specific because it depends on both wound width and sap velocity. However, our suggested threshold is within the magnitude of the threshold observed by Burgess et al. (2001). We have elaborated this justification in the text:

"The magnitude of the threshold chosen for the slope was taken from the modelled output of instantaneous ratios performed by Burgess et al. (2001), were low heat pulse velocities (5 cm h$^{-1}$) combined with a small wound width (0.17 cm) were shown to display a slope of 0.001. Due to our low heat pulse velocities ($< 15$ cm h$^{-1}$) and small probe diameter (0.12 cm), we expected the slope to be as close to 0.001 as possible. The specific threshold of 0.003 was decided upon inspection of the natural variability of the measurements and can be modified according to needle size and magnitude of the sap velocities. According to Burgess et al. (2001), higher values of slope (0.01) can be expected with greater wound width and higher velocities."

In relation to the referees' request for details regarding how the misalignment was estimated in figure 2, we included the exact equation in the method section:

"Each probe placement is calculated as:

$$x_2 = \sqrt{\left(4kt \ln \left(\frac{v_1}{v_2}\right) + 0.6^2\right)} \qquad [6]$$

Were $x_2$ is the incorrectly spaced probe, and 0.6 represents probe $x_1$, here assumed to be correctly spaced at 0.6 cm distance from the heater. This calculation is repeated for both probes. Two different heat pulse velocities, $V_1$ and $V_2$, are then derived (using equation 2 but with the assumption $y_1 = -y_2$) with the $x_1$ and $x_2$ obtained; and the final $V$ provided as their average."

We have also decided to include the misalignment from all 8 probes (Fig. 2).

[Figure]

[Figure]

Figure 2. The placements of the probes are shown as distance from the heater (cm). Probe placement was calculated once (solid lines) for the whole study period and compared to probe placement calculated varies times (solid circles with dotted lines) throughout the study period. Each point represents the probe position calculated during its respective zero-flow event.

3. I feel that the hydrological community and the readership of this Journal, would appreciate also some figures with the up-scaled transpiration estimates and the resulting biases do to probe misalignment, complementing the existing figures with the sap velocities and the results presented in L265-268.

We decided to combine the answer for this request with number 5, and have included a dataset expressed in transpiration (L tree$^{-1}$ h$^{-1}$) for each pine over 3 weeks towards the end of the study period.

4. The suggested time-depended correction accounts for two effects: probe misalignment and wounding effects. The current experimental design does not allow to disentangle the two. Therefore, the text should be revised so it is clear that the proposed correction addresses issues related to both wounding and probe misalignment.

In the text:

"The "perfect symmetry" assumption renders that HPR remains constant with time if heat pulse velocity ($V$), thermal diffusivity ($K$) and probe positions (in both, x and y directions) have negligible variations during the time following each heat pulse (Marshall, 1958). However, Burgess et al. (2001) demonstrate how empirical results initially differ from the ideal approach described by equation 3 due to blocking of xylem vessels and misalignment of sensors. However, the study concludes that the HPR converge asymptotically at least 60 seconds after the heat pulse release and, for at least 40 seconds more (until 100 seconds after the heat pulse release), which is when the HPR should be measured. Our study argues that a visual inspection of heat pulse velocities ($V$ in equations 2 and 3), does not necessarily give enough information to decide if measured values are a good representation of the sap flow. The method does not consider that random HPR can arise, which due to the sensitivity of the measurement, are likely to occur. On these premises, we have built a methodology to quality check sap flow measurements systematically by means of introducing a statistical analysis performed on the instantaneous heat pulse ratios acquired between 60 and 100 seconds after the heat pulse release. Hereafter, we will denote the averaged instantaneous heat pulse ratio between 60 and 100 seconds as HPR. The quality check consisted of establishing threshold values for relative standard deviation (RSD), statistically defined as the standard deviation divided by the mean, and the slope versus time, of the instantaneous heat pulse ratios. The statistical information obtained would account for any deterioration of the measurement. Burgess et al. (2001) proposed two separately methods to correct for wound and misalignment. The methods assume that errors arising from the wound inflicted by a sensor probe can be estimated using an empirical factor, whereas a misalignment of the probe needs to be calculated in situ. We propose a development of the misalignment correction method, while arguing that a statistical check of the HPR would detect a deterioration of the signal caused by a worsening of the wound. This would lead to a smaller sample mean and hence a higher RSD and was therefore chosen as a quality-check parameter along with the slope, which was a parameter proposed by Burgess et al. (2001)."

5. I suggest to include a comparison between sap velocities/transpiration estimates averaged throughout the study period/growing season as calculated with (i) no wounding correction, (ii) traditional (no time depended) corrections, and (iii) the presented time-depended correction. This would better emphasise/illustrate the advantages of this Technical note.

We appreciate the suggestion and agree that this would highlight the advantage of our study. We decided to include three weeks' worth of data towards the end of the study period to illustrate the biggest differences. This also highlighted a very small difference between the "non-correction" method and "time-dependent-correction" when the misalignment was small, or converged towards the ideal distance of 0.6 cm between the probes:

[Figure]

[Figure]

[Figure]

[Figure]

Figure 6. Timeline of transpiration estimates (L tree $^{-1}$ h$^{-1}$) for each sensor and each tree during 3 weeks at the end of the measuring period. Three estimations are shown; without any correction for misalignment of the probes (yellow line), one-time correction for misalignment of the probes (dark green), and time-dependent correction for misalignment of the probes (light green). The timeline represents diurnal measurements taken at a 30-minute interval.

Specific comments:

Abstract: the study location, tree species, number of instrumented trees, study period
should be clearly stated in the abstract.
L16: 'Whole-plant transpiration' reads redundant, just 'Transpiration' should be enough
here.
L17: and Hydrology.
L18: 'wide application range' and L19: 'ready automation': unclear what you mean
here. Please consider revising/rephrasing. Similar for 'data readings', I guess what
you mean here is the sap flow sensors can provide long-term measurements of sap
flow in tree stems with high temporal resolution (e.g., minutes, hours etc.).
L19: 'Several different': reads redundant. 'Several methods' or 'Different methods'
should be enough. L20: how the methods were adjusted to different climatic conditions?
Unclear statement. Maybe 'tested' instead of 'adjusted'?
L21: 'in the method', unclear to which method you are refereeing to, here. Please
rephrase/revise.
L21-22: if you focus only on the heat pulse method, then that is probably fine, but if you are referring to sap
flow methods in general, then additional sources of uncertainty
should be listed here, e.g., Granier's empirical parameters, zero-flow conditions, see
for example [ref]

We rephrased the abstract according to the specific comments proposed by the refree:

**"Abstract**
Transpiration is a crucial component in the hydrological cycle and a key parameter in many

disciplines like agriculture, forestry, ecology and hydrology. Sap flow measurements are one of the

most widely used methods to estimate whole-plant transpiration in woody species due to its

applicability in different environments and in a variety of species, as well as having the capacity of

continuous high temporal resolution measurements. Several methods have been developed and tested

under different climatic conditions. For low sap flow rates, the heat pulse ratio method has proven

most accurate. However, the scientific literature also identifies several sources of error for the method that needs to be accounted for; misalignment of the probes, wound to the xylem, thermal diffusivity and stem water content. This study aims to integrate probe misalignment as a function of time to improve measurements during long-term studies (> 3 months). Additionally, we propose a new set of statistical information to be recorded during the measurement period to use as a quality control for the heat ratio readings obtained from the sensors. Sap flow sensors were installed in four *Pinus halepensis*, in a coastal valley in South-Eastern Spain (39º57'45" N, 1º8'31" W) in a Mediterranean climate, for 20 months. We conclude that even when geometrical misalignments errors are small, the introduced corrections can generate an important shift in sap flow estimations. Relative standard deviation and the slope versus time of the instantaneous heat pulse ratio was used as quality indicators to conclude that the sensors showed no sign of deterioration after 20 months of deployment. Therefore, no general time limit can be decided for the longevity of the sensors but should rather be determined from individual performance over time."

We have further corrected for all the specific comments mentioned by the referee. In addition we have gone through the manuscript to make sure of the consistency of terms and expressions. In addition we would like to give a reply to some of the specific comments below.

L90-100: mentioned that you deployed eight sensors in total, two per tree. I found this information further below in the text, but this has to be very clear from the methods session.

We apologise for not using consistent terminology when referring to the sensors. There is one sap flow sensor per tree. Each sap flow sensor consists of two probes and one heater. When calculating the misalignment, we refer to each probe, of which there are 8. We have now declared a definition in material and methods:

"Each sap flow sensor consists of three needles: one heater and two thermocouples. We will refer to the thermocouples as probes, and when using the term "sensors" we refer to both probes and the heater"

L94-97: mention the specific depth where the thermocouples are located, and thus the heat velocity is measured. I found this information mentioned in a figure caption (L229) but has to be included in the methods description.

The depth of which the thermocouples are located is described in material and methods in the original paper (L94-97):

"The sensors were drilled into the uphill side of each tree trunk. Since *P. halepensis* has a higher sap flow average near the cambium with the flow steadily declining nearer to the heartwood (Cohen et al., 2008), sensors were installed at 20 mm depth below the cambium for average sap velocity rates, as estimated by Manrique-Alba (2017)."

The figure caption (L229) refers to the vertical distance between the heater and each of the thermocouples. This information is also included in materials and methods in the original manuscript (L98). However, we understand that this information can be interpreted as the depth, and we have therefore added to the caption:

"Figure 1. (A) Heat pulse ratio (HPR) throughout the measurement period in 30-minute intervals in tree number 1. Each HPR is an average of 41 instantaneous ratios corresponding to the temperature difference in two thermocouples at 0.6 cm up-and downstream from a heater probe at 0.2 cm depth:"

L138: you are referring to the raw heat velocities here I assume and not to sap flow measurements. Here and throughout the text clarify and use carefully and consistently terms such as heat velocity, sap velocity, and sap flux density.

We included a clarification in the specific phrase:

"On these premises, we have built a methodology utilising a quality check of systematic sap flow measurements by means of a statistical analysis performed on the instantaneous heat pulse ratio acquired between 60 and 100 seconds after the release of a heat pulse."

To me be more precise, we have gone through the whole text and decided to go away from the term sap velocity (cm h$^{-1}$) and use the term sap flow (cm$^3$ cm$^{-2}$ h$^{-1}$), denoting the sap volume flowing per square centimetre of sapwood per hour. This also makes it clearer to distinguish from heat pulse velocity.

---

## Author Response (AR1)

Dear Theresa Blume, thank you for handling our manuscript.

Below is a point-by-point response to the comments. Referees' comments are in black with our response in blue below each comment. The marked-up manuscript is found below the text.

Best regards,

Elisabeth K. Larsen with co-authors

**Referee number 1.**

The main issue I see with regards to the manuscript is that the authors could spend more attention on the variability of misalignment issues between installed trees and present in a clear way how they define zero flow conditions. Additionally, the implications of the correction could be more concretely quantified by showing time-series of daily water use, with and without the proposed correction. Finally, within the discussion there is space for further elaborating on the other issues related to the installment of these type of sensors. Clearly circumferential variability, wounding and other biases should be further investigated in the future.

We have gone through and corrected for the issues the referee points out. Because this is a technical note and we didn't want to increase the figures-to-text ratio too much, some of the additional figures are added in the supplementary materials. Figures are here shown below the text.

**Specific comments:**

Line 44-46: Please provide sources describing each of the method. Now readers cannot read related literature to explain the method.

We've included references in the text for each method mentioned, Line 40:

"There are a range of different approaches to sap flow measurements, and methods vary between heat dissipation (HD), (Granier, 1987; Lu et al., 2004), steam heat balance (SHB), trunk segment heat balance (THB), (Smith and Allen, 1996), and heat pulse velocity (HPV), (Marshall, 1958). However, they are all based on tracing heat within the xylem (Burgess et al., 2001; Davis et al., 2012; Forster, 2017)."

Line 56-59: See also: Steppe et al. 2010 A comparison of sap flux density using thermal dissipation, heat pulse velocity. This is a relevant study which addresses the offset of sap flow methods from gravimetric measurements.

Thank you for providing the reference, even if the paper is comparing the overall heat pulse velocity technique rather than the specific heat ratio method, we found it very relevant and decided to include the reference, line 50:

"By accounting for these sources of error and additionally estimating the stem moisture content and radial variability, the heat ratio method (HRM) has been evaluated the heat pulse velocity method (HPV) with highest accuracy, although with a tendency of underestimating transpiration values (Forster, 2017), an error that is shown to increase with higher sap flow values (Steppe et al., 2010)."

The paper also made us aware of an error of concept; using the term "sap velocity" and the unit cm h -1 is not precise, and we have therefore changed it throughout the paper to "sap flow" per unit of sap wood ( $\text{cm}^3 \text{ cm}^{-2} \text{ h}^{-1}$ ).

Figure 1: It would be good if there would be a zoom in panel where you can see more detail on the patterns. The current point cloud does not give the reader a full idea on the diurnal quality of the data.

As suggested by the referee, we have included a zoomed in panel in Fig.1, representing one week of data for the HRP, slope and RSD, marked A1, B1 and C1. We agree that it enhance the understanding of the filtration as it

gives the reader the diurnal pattern of the data and therefore demonstrates the range of values that has been filtered out.

Line 220: Please help the reader to understand what RSD is. I went back to the methods to check, yet this part of the text should be understandable on its own.

We included a definition for the RSD also in this part of the text so the reader doesn't need to go back in the text to check, line 230:

"Therefore, the quality of the measurements was indicated by calculating the RSD, the relative standard deviation divided by the sample mean, and the slope versus time, for each HPR."

Line 237-238: It would be good to understand when and why there is a limited amount of periods with zeroflow conditions. Additionally, it is not clear how these period where exactly defined. It would help to include an appendix figures which details these periods and the underlying environmental conditions.

As suggested by the referee, a definition of zero-flow conditions was included. The definition now references five additional figures in the appendix, showing the relative extractable water and VPD along with the heat pulse ratios during the estimated zero-flow events, line 250:

"Zero-flow conditions were assumed at night (22:00-03:00h solar time) during days when relative extractable water, REW > 0.75, and vapor pressure deficit (VPD) was close to zero (Fig. S2). Multiple readings were used to produce an average of each event. These conditions were limited to five occasions during our study period."

An additional paragraph was included in the method section due to the inclusion of environmental data, line 95-100:

**"Environmental conditions**

Air relative humidity (%) and air temperature (°C) were registered every 30-minute (U23 Pro V2, Onset Computer Corporation, USA). Precipitation was registered using a rain gauge with 0.2 mm resolution (RG3-M, Onset Compute, USA). Three soil moisture probes were inserted at 20-25 cm depth to register soil water content, SWC, (S-SND.M005, Onset Computer Corporation, USA), using a datalogger for logging specifications (HOBO Micro Station, USA). To assume periods of zero-flow events, relative extractable water (REW) was calculated using the method of Bréda et al. (1995):

$$REW = \frac{(\theta_t - \theta_{min})}{(\theta_{max} - \theta_{min})}$$
[1]

Were  $\theta_t$  is the registered SWC,  $\theta_{min}$  is the minimum SWC observed during the measurement period, and  $\theta_{max}$  is the SWC at field capacity. When values of REW surpassed 1 they were converted to 1 according to Granier et al. (2000)."

Additionally, now the displacement is provided for one tree and the average of all trees and sensors. Yet, it would be good to see whether there are differences between the sensors themselves. Would the authors be able to provide the change presented in Figure 2 for each sensor?

On request from the referee, we decided to include the change described in Figure 2 for each sensor. We also included a better definition of the sensor to clarify that each sensor consists of three needles, two probes and one heater. When referring to misalignment we refer to both probes in a sensor.

Line 85:

"Each sap flow sensor consists of three needles: one heater and two thermocouples. We will refer to the thermocouples as probes, and when using the term "sensors" we refer to both probes and the heater."

Regarding the misalignment, line 250:

"The outputs indicate a clear shift of placement in each of the probes over time, here denoted as x1 and x2 in each pine (Fig. 2). The eight probes, two per tree, all deviated from the ideal of 0.6 cm. Probe x2 in pine number 1 had the highest inaccuracy with an initial value close to 0.3 cm".

Line 263-265: It would be interesting to see temporally what the difference are in daily water use (L d-1). This will clarify if the offset due to misalignment is progressively getting worse or whether, in these species, the impact is not that bad. Additionally, when presenting these numbers, it is critical that the standard deviation is also provided for these estimates.

The authors originally didn't include the difference of daily water use  $(L d^{-1})$  because this introduces the errors related to upscaling to whole tree transpiration, whereas we wanted to focus on the correction of the point measurements. However, we chose to include three weeks of data towards the end of the measurement period, also including the transpiration values without any misalignment correction (Fig. 6). This also highlighted a very small difference between the "non-correction" method and "time-dependent-correction" when the misalignment was small or converged towards the ideal distance of 0.6 cm between the probes. In the original manuscript, Fig. 5 represents how the difference between the two methods changes in time and is meant to give an indication of how the misalignment estimation is getting progressively worse if the misalignment is measured only once at the beginning of a measurement campaign. Line 266-268 in the original manuscript shows the overall difference between using the correction during the entire study period of 20 months. Standard deviation was provided for the estimates in table 1.

Line 296-299: Indeed, there could be a reduction in the amplitude due to wounding effects or other changes within the stem. Did the authors analyse whether they would see a reduction in the amplitude over time? It would be important to make this test as the data is available.

We have taken the suggestion of the referee into consideration and included the test results in the paper and the figures in the supplementary material, line 320-325:

"Due to the variation in rainfall between the two years, SWC differed significantly on similar calendar dates with the exception of six days in June (9 – 15). When comparing the relationship of sap flow versus VPD on these days, there was an increase in the slope the second year: 2.3 for 2017 and 2.9 for 2018. This, we attributed to an overall weaker correlation between VPD and sap flow in 2017; R 2 = 0.4\* in 2017 versus R 2 = 0.6\* in 2018, due to higher values of VPD in 2017 (S3). Within the normal variability of the measured data we concluded that the sap flow values had not decreased in the second year, when compared to the first year, under similar environmental conditions."

Line 316-318: This is indeed a valid point, yet I would propose that the authors would elaborate on the fact that reinstalling sensors along the stem will introduce change due to circumferential variability in the stem. This could be critical when generating continuous series of sap flow over the long term.

We appreciate the referee pointing this out, as it is an important point to make which we now have included in the text, line 350:

"However, as pointed out by Moore et al. (2010), re-installing the sensors might create a shift in the data due to spatial variation within the tree. Leaving the same sensors in the tree throughout the study period avoids this problem and enables the study to focus on the intrinsic factors affecting the sap flow rates"

Also, do the authors think these results found on conifers are universal for all types of species? I would have expected a short discussion to clarify to the reader why these findings could be of general value to the application of the method.

We agree that it's an important addition to the discussion and we have therefore included it in the text, line 390 - 340:

"The time-dependent correction method could be useful with any species already tested with the HRM, where misalignment of probes can create a source of error and sensors are installed over a longer period. Specifically, where the movement of the wood might cause further displacement of the sensor."

**Response to referee number 2:**

Overview: Larsen et al. in this Technical note address two important issues of sap flow measurements with the heat pulse method, namely (i) data filtering/quality control of the raw heat pulse records, and (ii) errors due to misalignment of the sap flow probes. The authors suggest some statistical thresholds/filters to be applied in the raw heat pulse ratios for data cleaning and present a time-dependent correction to account for probe misalignment. Moreover, they demonstrate the importance of such uncertainties for robust transpiration estimates, by presenting sap velocity and transpiration estimates with and without applying the proposed correction. I find the study topical and of interest for the scientific community working on transpiration estimates with the heat pulse sap flow method. However, I feel that the manuscript needs major revisions to better present the motivation and rationale of the study, the developed methods, and the broader implications of the obtained results.

**To better present the motivation of the study we have made some changes in the text, line 55-60:**

"Previous studies have suggested additional solutions for probe misalignment (Ren et al., 2017), for determining thermal diffusivity (Vandegehuchte and Steppe, 2012), and correcting for heterogeneous heat capacity in wood (Becker and Edwards, 1999). However, there is no recent recommendation for how long newly deployed sap flow sensors can be employed. Some studies have shown how sensor probes inserted into the xylem can dampen the signal due to blocking or destruction of vessels (Moore et al., 2010; Wiedemann et al., 2016). One way to account for changes over time has been to reinstall sensors throughout the study period (Moore et al. 2010), however there is little information to be found on the exact interval for which this needs to happen (Vandegehuchte and Steppe, 2013), and for continuous measurements this will interrupt the dataset (Moore et al. 2010). Therefore, we aim to find a

quality indication that can ensure that the readings don't deteriorate over time, or if they do, that it would be detectable. Attention should be given to check the accuracy of the heat pulse ratio itself, in which the rest of the methodology is built on. In addition, to allow for sensors to be employed over longer periods it's necessary to develop a dynamic probe misalignment correction method due to observed change in probe position over time."

The text needs significant editorial improvements to eliminate vague/unclear wording and grammatical errors. Moreover, several parts of the text (including the abstract) need to be revised/rephrased/rewritten to improve the clarity of the text and better communicate the design of the study, results, discussion and conclusions. I have highlighted few specific points below (see Specific comments), yet several other cases exist throughout the manuscript.

We have corrected specific comments and rewritten phrases in the abstract, methods, discussion and conclusion. We hope this has led to more clarity and improved the communication of our study. Specifically we rewrote the conclusion, line 355-365.

The methods need to be revised and clarified. In some parts, there are inconsistencies and it is hard to follow. Sometimes the authors refer to V as sap velocity (L130) and other times as heat pulse velocity (L113, L135). Please clarify and use consistently the terms/variables/abbreviations throughout the manuscript. Also, the selected thresholds (L153-161) for the raw data filtering need to be better justified, since at the moment seem quite arbitrary or could be interpreted as case-specific. Also, the data from all eight sensors (or averages across trees, since two sensors per tree were deployed) should be presented, either in the main text, or in the supplementary material. Apart from Fig 5, all figures illustrate data from a single sensor. In addition, more details should be provided in the methods session on how the positions of sensor misalignment were estimated in Fig 2 (and the misalignment for all eight sensors would be interesting to be illustrated, too).

Inconsistencies highlighted by the referee has been addressed and corrected for. Further inconsistencies or vague formulations has been checked throughout the manuscript. We argue that the selected threshold needs to be case-specific because it depends on both wound width and sap velocity. However, our suggested threshold is within the magnitude of the threshold observed by Burgess et al. (2001). We have elaborated this justification in the text, line 165-170:

"The magnitude of the threshold chosen for the slope was taken from the modelled output of instantaneous ratios performed by Burgess et al. (2001), were low sap flow velocities (5 cm h-1) combined with a small wound width (0.17 cm) were shown to display a slope of 0.001. Due to our low sap flow velocities (< 15 cm h-1) and small probe diameter (0.12 cm), we expected the slope to be as close to 0.001 as possible. The specific threshold of 0.003 was decided upon inspection of the natural variability of the measurements and can be modified according to needle size and magnitude of the sap velocities. According to Burgess et al. (2001), higher values of slope (0.01) can be expected with greater wound width and higher velocities."

In relation to the referees' request for details regarding how the misalignment was estimated in figure 2, we included the exact equation in the method section as equation 6. We have also decided to include the misalignment from all 8 probes (Fig. 2).

I feel that the hydrological community and the readership of this Journal, would appreciate also some figures with the up-scaled transpiration estimates and the resulting biases do to probe misalignment, complementing the existing figures with the sap velocities and the results presented in L265-268.

We decided to combine the answer for this request with number the request for comparison between sap flow with and without corrections. A comparison is now included as figure 6 expressed as transpiration estimates (L tree -1 h-1) for each pine over a three-weeks period at the end of the study.

The suggested time-depended correction accounts for two effects: probe misalignment and wounding effects. The current experimental design does not allow to disentangle the two. Therefore, the text should be revised so it is clear that the proposed correction addresses issues related to both wounding and probe misalignment.

**The text was revised accordingly, line 145-150.**

I suggest to include a comparison between sap velocities/transpiration estimates averaged throughout the study period/growing season as calculated with (i) no wounding correction, (ii) traditional (no time depended) corrections, and (iii) the presented time-depended correction. This would better emphasise/illustrate the advantages of this Technical note.

We appreciate the suggestion and agree that this would highlight the advantage of our study. We decided to include three weeks' worth of data towards the end of the study period to illustrate the biggest differences. This also highlighted a very small difference between the "non-correction" method and "time-dependent correction" when the misalignment was small, or converged towards the ideal distance of 0.6 cm between the probes, as Figure 6 in text.

Specific comments: Abstract: the study location, tree species, number of instrumented trees, study period should be clearly stated in the abstract. L16: 'Whole-plant transpiration' reads redundant, just 'Transpiration' should be enough here. L17: and Hydrology. L18: 'wide application range' and L19: 'ready automation': unclear what you mean here. Please consider revising/rephrasing. Similar for 'data readings', I guess what you mean here is the sap flow sensors can provide long-term measurements of sap flow in tree stems with high temporal resolution (e.g., minutes, hours etc.). L19: 'Several different': reads redundant. 'Several methods' or 'Different methods' should be enough. Le20: how the methods were adjusted to different climatic conditions? Unclear statement. Maybe 'tested' instead of 'adjusted'? L21: 'in the method', unclear to which method you are refereeing to, here. Please rephrase/revise. L21-22: if you focus only on the heat pulse method, then that is probably fine, but if you are referring to sap flow methods in general, then additional sources of uncertainty should be listed here, e.g., Granier's empirical parameters, zero-flow conditions.

We rephrased the abstract according to the specific comments proposed by the referee.

We have further corrected for all the specific comments mentioned by the referee. In addition, we have gone through the manuscript to make sure of the consistency of terms and expressions.

L90-100: mentioned that you deployed eight sensors in total, two per tree. I found this information further below in the text, but this has to be very clear from the methods session.

We apologise for not using consistent terminology when referring to the sensors. There is one sap flow sensor per tree. Each sap flow sensor consists of two probes and one heater. When calculating the misalignment, we refer to each probe, of which there are eight. We have now declared a definition in material and methods, 85:

"Each sap flow sensor consists of three needles: one heater and two thermocouples. We will refer to the thermocouples as probes, and when using the term "sensors" we refer to both probes and the heater"

L94-97: mention the specific depth where the thermocouples are located, and thus the heat velocity is measured. I found this information mentioned in a figure caption (L229) but has to be included in the methods description.

The depth of which the thermocouples are located is described in material and methods in the original paper (L94-97):

"The sensors were drilled into the uphill side of each tree trunk. Since P. halepensis has a higher sap flow average near the cambium with the flow steadily declining nearer to the heartwood (Cohen et al., 2008), sensors were installed at 20 mm depth below the cambium for average sap velocity rates, as estimated by Manrique-Alba (2017)."

The figure caption (L229) refers to the vertical distance between the heater and each of the thermocouples. This information is also included in materials and methods in the original manuscript (L98). However, we understand that this information can be interpreted as the depth, and we have therefore added to the caption:

"Figure 1. (A) Heat pulse ratio (HPR) throughout the measurement period in 30-minute intervals in tree number 1. Each HPR is an average of 41 instantaneous ratios corresponding to the temperature difference in two thermocouples at 0.6 cm up-and downstream from a heater probe at 0.2 cm depth:"

L138: you are referring to the raw heat velocities here I assume and not to sap flow measurements. Here and throughout the text clarify and use carefully and consistently terms such as heat velocity, sap velocity, and sap flux density.

We included a clarification in the specific phrase, line 140:

"On these premises, we have built a methodology utilising a quality check of systematic sap flow measurements by means of a statistical analysis performed on the instantaneous heat pulse ratio acquired between 60 and 100 seconds after the release of a heat pulse."

To me be more precise, we have gone through the whole text and decided to go away from the term sap velocity (cm h-1) and use the term sap flow (cm3 cm-2 h-1), denoting the sap volume flowing per square centimetre of sapwood per hour. This also makes it clearer to distinguish from heat pulse velocity.

**Additional changes:**

- After adding more figures as suggested by both the referees, we decided that figure number 3 in the original manuscript was superfluous. We realised that the figure, showing heat pulse velocities from the whole study period, gives an unclear picture of the actual baseline due to the large amount of data points. The zoomed-in panels of the same data (Figure 4 in the original manuscript), gives the same information for one week at the time.
- Another paragraph was added in the result section to complement the graph showing the transpiration values, and the differences between the values in the new manuscript (line 295-100).

**Technical note on long-term probe misalignment and proposed**

**quality control using the heat pulse method for transpiration**

**estimations**

Elisabeth K. Larsen1,3, Jose Luis Palau1, Jose Antonio Valiente1, Esteban Chirino2 and Juan Bellot3,4.

[revised manuscript text omitted]

**3 Materials and method**

**3.1 Field site**

This study was carried out in the Turia river basin, Eastern Spain (39°57'45" N 1°8'31" W), in a Mediterranean climate. Average annual rainfall is 475 mm, average annual maximum and minimum temperature is 15.5 °C and 4.4 °C respectively. Sap flow sensors were installed in four pine trees (*Pinus halepensis* Mill.) according to the heat ratio method (HRM, Burgess et al., 2001). Each sap flow sensor consists of three needles; one heater and two thermocouples. We will refer to the thermocouples as probes, and when using the term "sensors" we refer to both probes and the heater. Three needles, one heater and two The sensors thermocouples (0.13 cm × 4 cm), were drilled into the uphill side of each tree trunk. Since *P. halepensis* has a higher sap velocity average near the cambium with the velocity steady declining nearer to the heartwood (Cohen et al., 2008), sensors were installed at 20 mm depth below the cambium for average sap velocity rates, as estimated by Manrique-Alba (2017). A metal plate was used as guide to ensureassure a 0.6 cm spacing between the drilling holes. The *P. halepensis*-selected pines had an average 
[revised manuscript text omitted]

---

## Author Response (AR2)

Dear Theresa Blume, thank you for your handling of this manuscript and your feedback. Below you will find a point-by-point answer to all suggestions with the referees' comments in black and authors comment in blue.

Referee number 1:

Thank you for staying with this manuscript and revising it for the second time. We are glad to know you are happy with the corrections made to the manuscript and the additional figures and table that you highlight.

Line 35: Consider including: 1) Peters et al. 2018 2) Flo et al. 2019.
Line 35: I thank the referee for important and relevant additions to the literature list. The Peters et al., 2018 and Flo et al., 2019 was added.

Line 37: For SHB one can use: Langensiepen M, Kupisch M, Graf A, Schmidt M, Ewert F. 2014. Improving the stem heat balance method for determining sap-flow in wheat. Agricultural and Forest Meteorology 186:34-42.
L40: Thank you for the suggestion, Langensiepen et al., 2014 was added for the SHB method as suggested by the referee.

Line 30: The manuscript from Jasechko et al. 2013 has received some criticism (http://dx.doi.org/10.1038/nature12925), consider citing: Schlesinger WH, Jasechko S. 2014. Transpiration in the global water cycle. Agricultural and Forest Meteorology 189-190:115-117.
L31: The article of Jasechko et al., 2013 was replaced with the one from Schelsinger and Jasechko, 2014. We found that the same statement still held, only they had adjusted the values of transpiration/evapotranspiration ratio in the second paper.

Line 71: Why specifically 3 months? Is this the time in which the authors do not expect any wounding effect to develop? Or is this the time which is commonly used for monitoring? Please clarify.
This was clarified in the manuscript accordingly:
> L74: "The adjustment proposed here, which is built on the calculations of Burgess et al. (2001), is necessary when monitoring transpiration continuously for more than 3 months due to observed shift in probe placement from one season to another (Fig. 4, 6)."

Line 76: "…probe misalignment correction for the HRM.." Remove one dot at the end of this sentence please.
Removed.

Line 86-87: Please explain why you drilled the sensors into the uphill side of the tree
We clarified this by adding:
> L96-97: To a establish a criterion that unified the effect of the slope on all trees, with the upper part facilitating the retention of water in the soil, all the sensors were drilled into the uphill side of each tree trunk.

Line 273: "… 0.29 (cm cm3 cm-2 h-1)". I presume the authors should remove cm?
Yes, removed.

Lines 274: "…(Table 1, Fig. 4) In terms of…". A dot is missing after "Fig. 4)".
Added.

We thank referee number 2, Miriam Coenders-Gerrits, for taking an interest in our manuscript and for the valuable feedback. Comments are answered point-by-point below.

- P1L30: Jasechko later corrected his global estimate for transpiration (Schlesinger & Jasechko 10.1016/j.agrformet.2014.01.011)
L31: Yes, this was suggested by both referees and has been changed.

- P2L46: I would elaborate on what you mean by probe misalignment. Maybe a schematic drawing would help.
L55: We appreciate the suggestion and have included a schematic drawing of the probe misalignment (Fig. 1), which we hope improves the understanding of the method.

- P2L74: methodological paper=> technical note
Changed.

- P2L76: remove one dot after HRM
Removed.

- P2L83: unit of annual rainfall is mm/YEAR
Yes, thank you.

- P3L114-122: add units.
L132-140: Units added.

- P4L124: an image would help to understand your coordinate system.
L140: We included a photo of the probes showing the coordinate system we describe in the text. Hopefully giving the reader a good understanding of the positions.

- P5L189: what is both x1 and x2 are misaligned? Does your method work then? Please explain and/or elaborate on this.
We added to an explanation for this in the text L208-210:

"Usually both $x_1$ and $x_2$ are slightly misaligned, which means that each correction step will assume an error, that the other probe is correctly placed. However, taking the average of the misalignment calculated from each probe, both misalignments will be accounted for (Burgess et al., 2001)."

- P7L256" remove "average". This is already said in L255.
Thank you.

Other corrections:
Note that we have added an addition to equation 7, it is explained in the text accordingly:

[revised manuscript text omitted]

---

## Editor Decision (ED2)

Dear Theresa Blume, thank you for spending time on this manuscript and for valuable feedback. We have, has requested, gone through it thoroughly and corrected for typos and unclarities. Below you will find point-by-point answers to your comments and a marked-up version of the manuscript. We look forward to your response.

L21-22: This is confusing. Do you mean that even when misalignment is small it can cause significant shifts/errors in sap flow estimation? Here it sounds like the misalignment errors are small but in contrast your corrections cause big shifts.

Yes, this is what we mean, that the difference in the output can be big although the misplacement is quite small. We have rephrased to make it read more clearly:

L22-23: "We conclude that even small geometrical misalignments in probe placement can create a significant error in sap flow estimations".

L25: Explain what this is.

We have explained this in the text (L149 and L161), and decided to not include this term in abstract, as it can cause confusion as pointed out by the editor.

L60: Did you contact the journal/authors (whoever has the copyright) for the right to reproduce these figures?

Yes, we have the permission from the authors and the journal (open access). We included 'with permission' in the figure legend.

L70: make clear what this is

We decided it would be best to introduce this term in the method section, and have therefore changed the phrase to (L69):

"Attention should be given to check the accuracy of the raw readings, in which the rest of the methodology is built on"

L96-L98: this is confusing/not clear. Do you mean: "As the wood differs between the upslope and downslope sides of the trees, we decided to consistently install all sensors into the upslope side."
I am not really following your statement about the water retention in the soil. Did you measure this? Is this relevant here? Another common criterion is to install the sensors

facing north to decrease solar radiation on the sensors. Were your sensors covered with radiation shields?

We understand that this wasn't clear. We have now emphasised that we are talking about the slope of the ground, and therefore it is relevant with the retention of water in the soil. It was not measured but used to set a criterion for all sensors when installed (L95-L97):

"To establish a criterion that unified the effect of the slope of the ground, ==with the upper part facilitating the retention of water in the soil,== all the sensors were drilled into the uphill side of each tree trunk. All sensors were covered with radiation shields."

L125: Do you mean v1/v2? I do not think this is the common term for this

The v1/v2 ratio is an average ratio of multiple instantaneous ratios taken before the release of a heat pulse and between 60 and 100 seconds. It does not have a common term because it is not usually referred to as it is more common to refer to the averaged ratio (v1/v2) instead. However, the v1/v1 is not a precise term because averages are taken from however many heat ratios one chooses to use. In our case this is 41 (Fig. S1).

In the script (L125-133):

"Marshal (1958) parameterised the heat pulse velocity (V) in the HRM as a function of time (s) following a heat pulse, and the instantaneous ratio of the increase in temperature (ºC) (from the temperature prior to the release of the thermal pulse) at the downstream ($v_1$) and upstream ($v_2$) from a line heater

$$= \frac{4Kt \ln\left(\frac{v_1}{v_2}\right) - (x_2^2 + y_2^2) + (x_1^2 + y_1^2)}{2t(x_1 - x_2)} \qquad [2]$$

where $K$ is the thermal diffusivity ($cm^2 \, s^{-1}$), $t$ is the time passed from the release of thermal pulse in seconds, $(x_1, y_1)$ and $(x_2, y_2)$ are the relative positions (cm) of the thermocouples to the heater (considering x-axis along the xylem and y-axis in the perpendicular direction) (Fig. 2), and $v_1$, $v_2$ representing the temperature increases following the heat release".

L139: it is a bit confusing that your x and y axes are the opposite of the standard x = horizontal and y= vertical axes

Yes, we agree that this can seem confusing. We also included an x in figure 1 to make sure that we are talking about the vertical axis when referring to x. For figure 2 we wrote (L136-137):

"The x-axis represents the vertical distance from the heater, while the y-axis accounts for horizontal probe alignment from the heater".

L142: if sensors are installed vertically then y1=y2?

We have included a clarification in the sentence (L140):

 "If probes are installed symmetrically, horizontally and vertically, in line with the heater, x = x1 = -x2 and y1 = y2=0."

L146: Make clear what this refers to.  This is x1, right? Positive distance of heater to probe?

We have added the description for clarity (L145-L147):

"where $\phi$ is a priori, a constant only depending on the placement of the probes and on the thermal diffusivity of both the xylem and the material used in the sensors, and x, a positive magnitude representing the distance from the heater element for the two probes".

Combined explanations:

L150: Explain what this is, L159: why instantaneous? L180: Not clear. What do you mean here?

Clarified (L149):

"The $\frac{v_1}{v_2}$ is an average of multiple instantaneous heat pulse ratios (HPR), one for each second of which the temperature is registered, at 60 to 100 seconds after the heat pulse release.".

Combined explanations: L181: is this correct? L263, Figure 3C: why do we have negative slopes? Shouldn't they be positive?

The slope needs to be as close to 0 as possible, it does not mean it has to be positive. However, it can be argued that the threshold we sat of |slope – median (slope)| < 0.003 s$^{-1}$ can vary depending on sensor performance. We clarified in the text as (L180):

"Deviations from an idealized slope of zero, positive or negative, means that the HPR is not being constant with time".

L183: Instantaneous ratios are not mentioned in Burgess et al.

Burgess et al. (2001) uses the $\frac{v_1}{v_2}$, but displays a figure with the relationship between this ratio and time in seconds. Each of those ratios is what we term instantaneous here, to go into detail of its slope and relative standard deviation.

L190-191: This is what you call HPR, or not?

Yes, we simplified accordingly (L192):

"the V1/V2 would be equal to 1 (v1 = v2 in equation 3)".

L203: or y1=y2

Yes! thank you for pointing this out.

Fig3A: but doesn't this figure show all the data, not just the night-time, wet conditions where we assume zero flow?

Yes, it is correct that the figure 3A includes all the data, but the baseline for the whole dataset should have been close to 1 and not somewhere between 1.4 and 1.5 as seen here.

L256: I don't understand this part. Please explain. why are these values eliminated?

Because the $\frac{v_1}{v_2}$ is an average of multiple instantaneous HPRs, the slope of each HPR will demonstrate how close they are to zero, which is what the slope should be if the sensors are perfectly symmetrical.

In the text it's explained accordingly (L183-186):

"The magnitude of the threshold chosen for the slope was taken from a modelled output of $\frac{v_1}{v_2}$ ratios performed by Burgess et al. (2001), were low sap flow velocities (5 cm h-1) combined with a small wound width (0.17 cm) were shown to display a slope of 0.001."

Your axis labels only show 0.01, we need more digits here!

Corrected, thank you.

L260: Is this a week were you assume that we have zero flow at night? Then state that.

No, it is not. I it is a zoomed in panel of a normal week of data. We included:
"Zoomed in panel of HPR data for one week of measurements to better see the diurnal pattern".

L265: You should show this in the plot - give the median and clearly indicate the areas on the plot with the erroneous data.

The read lines indicate the erroneous data. All data below or below these lines are filtered out.

L265: "Red line indicates threshold used for the quality control for this particular sensor. All data above or below these lines are filtered out as they did not correspond to HPR with |slope – median (slope)| < 0.003 s-1. Median slope for this sensor was -0.004"

Table1 and figure 6: Why not give the dates here? Or month and year?

We included the dates in the table description. Adding it directly to the header of the table would alter its format. Same for figure 6.

what do you mean by this?

L333-L336: "Because the HPR is an average of multiple instantaneous temperature ratios, it is difficult to ensure its accuracy only by visual inspection of the averaged output, without further statistical information. For each heat pulse measurement sequence, we suggest RSD and slope from linear regression of the instantaneous ratios versus time, to filter out random ratios and to ensure the quality of the data"

Combined explanations:

L337: What do you mean by this? erroneous? uncertain? Limited is too vague. L338: Not clear. This sentence is incomplete.

We have specified in the text accordingly:

"The HRM only measure values below 45 cm h-1, (Forster, 2017), due to a maximum ratio of v1/v2 = 20, for when it can be assumed accurately (Burgess et al., 2001; Marshall, 1958). Because our dataset showed no higher velocities than 8 cm h-1, it was not initially considered a limitation for the measurements.".

L347: S3 corrected to Fig. S3.

L384-L385: Deleted.

Please find the marked-up manuscript version below.

[revised manuscript text omitted]

[Figure]

Figure 2. Sensor placements, from top: thermocouple, heater, thermocouple, in an Aleppo pine. The x-axis represents the vertical distance from the heater, while the y-axis accounts for horizontal probe alignment from the heater. The axes should cross where the heater is inserted but was moved for a clearer view of the coordinates.

If probes are installed symmetrically, horizontally and vertically, in line with the heater, $x = x_1 = -x_2$ and $y_1 = -y_2 = 0$, equation 2 simplifies into a function that is not dependent on time:

$$V = \phi \ln(\frac{v_1}{v_2})$$

$$\phi = \frac{K}{x}$$

[3]

where $\phi$ is a priori, a constant only depending on the placement of the probes and on the thermal diffusivity of both the xylem and the material used in the sensors, and x, a positive magnitude representing the distance from the heater element for the two probes

The $\frac{v_1}{v_2}$ is an average of multiple instantaneous heat pulse ratios (HPR), one for each second the temperature was registered for, from 60 to 100 seconds after the heat pulse release. 
[revised manuscript text omitted]

---

## Author Response (AR3)

Dear Theresa Blume, thank you for spending time on this manuscript and for valuable feedback. We have, has requested, gone through it thoroughly and corrected for typos and unclarities. Below you will find point-by-point answers to your comments and a marked-up version of the manuscript. We look forward to your response.

L21-22: This is confusing. Do you mean that even when misalignment is small it can cause significant shifts/errors in sap flow estimation? Here it sounds like the misalignment errors are small but in contrast your corrections cause big shifts.

Yes, this is what we mean, that the difference in the output can be big although the misplacement is quite small. We have rephrased to make it read more clearly:

L22-23: "We conclude that even small geometrical misalignments in probe placement can create a significant error in sap flow estimations".

L25: Explain what this is.

We have explained this in the text (L149 and L161), and decided to not include this term in abstract, as it can cause confusion as pointed out by the editor.

L60: Did you contact the journal/authors (whoever has the copyright) for the right to reproduce these figures?

Yes, we have the permission from the authors and the journal (open access). We included 'with permission' in the figure legend.

L70: make clear what this is

We decided it would be best to introduce this term in the method section, and have therefore changed the phrase to (L69):

"Attention should be given to check the accuracy of the raw readings, in which the rest of the methodology is built on"

L96-L98: this is confusing/not clear. Do you mean: "As the wood differs between the upslope and downslope sides of the trees, we decided to consistently install all sensors into the upslope side."
I am not really following your statement about the water retention in the soil. Did you measure this? Is this relevant here? Another common criterion is to install the sensors

facing north to decrease solar radiation on the sensors. Were your sensors covered with radiation shields?

We understand that this wasn't clear. We have now emphasised that we are talking about the slope of the ground, and therefore it is relevant with the retention of water in the soil. It was not measured but used to set a criterion for all sensors when installed (L95-L97):

"To establish a criterion that unified the effect of the slope of the ground, with the upper part facilitating the retention of water in the soil, all the sensors were drilled into the uphill side of each tree trunk. All sensors were covered with radiation shields."

L125: Do you mean v1/v2? I do not think this is the common term for this

The v1/v2 ratio is an average ratio of multiple instantaneous ratios taken before the release of a heat pulse and between 60 and 100 seconds. It does not have a common term because it is not usually referred to as it is more common to refer to the averaged ratio (v1/v2) instead. However, the v1/v1 is not a precise term because averages are taken from however many heat ratios one chooses to use. In our case this is 41 (Fig. S1).

In the script (L125-133):

"Marshal (1958) parameterised the heat pulse velocity (V) in the HRM as a function of time (s) following a heat pulse, and the instantaneous ratio of the increase in temperature (°C) (from the temperature prior to the release of the thermal pulse) at the downstream ($v_1$) and upstream ($v_2$) from a line heater

$$= \frac{4Kt \ln\left(\frac{v_1}{v_2}\right) - (x_2^2 + y_2^2) + (x_1^2 + y_1^2)}{2t(x_1 - x_2)} \qquad [2]$$

where $K$ is the thermal diffusivity (cm$^2$ s$^{-1}$), $t$ is the time passed from the release of thermal pulse in seconds, $(x_1, y_1)$ and $(x_2, y_2)$ are the relative positions (cm) of the thermocouples to the heater (considering x-axis along the xylem and y-axis in the perpendicular direction) (Fig. 2), and $v_1$, $v_2$ representing the temperature increases following the heat release".

L139: it is a bit confusing that your x and y axes are the opposite of the standard x = horizontal and y= vertical axes

Yes, we agree that this can seem confusing. We also included an x in figure 1 to make sure that we are talking about the vertical axis when referring to x. For figure 2 we wrote (L136-137):

"The x-axis represents the vertical distance from the heater, while the y-axis accounts for horizontal probe alignment from the heater".

L142: if sensors are installed vertically then y1=y2?

We have included a clarification in the sentence (L140):

"If probes are installed symmetrically, horizontally and vertically, in line with the heater, x = x1 = -x2 and y1 = y2=0."

L146: Make clear what this refers to. This is x1, right? Positive distance of heater to probe?

We have added the description for clarity (L145-L147):

"where $\phi$ is a priori, a constant only depending on the placement of the probes and on the thermal diffusivity of both the xylem and the material used in the sensors, and x, a positive magnitude representing the distance from the heater element for the two probes".

Combined explanations:

L150: Explain what this is, L159: why instantaneous? L180: Not clear. What do you mean here?

Clarified (L149):

"The $\frac{v_1}{v_2}$ is an average of multiple instantaneous heat pulse ratios (HPR), one for each second of which the temperature is registered, at 60 to 100 seconds after the heat pulse release.".

Combined explanations: L181: is this correct? L263, Figure 3C: why do we have negative slopes? Shouldn't they be positive?

The slope needs to be as close to 0 as possible, it does not mean it has to be positive. However, it can be argued that the threshold we sat of |slope – median (slope)| < 0.003 s$^{-1}$ can vary depending on sensor performance. We clarified in the text as (L180):

"Deviations from an idealized slope of zero, positive or negative, means that the HPR is not being constant with time".

L183: Instantaneous ratios are not mentioned in Burgess et al.

Burgess et al. (2001) uses the $\frac{v_1}{v_2}$, but displays a figure with the relationship between this ratio and time in seconds. Each of those ratios is what we term instantaneous here, to go into detail of its slope and relative standard deviation.

L190-191: This is what you call HPR, or not?

Yes, we simplified accordingly (L192):

"the V1/V2 would be equal to 1 (v1 = v2 in equation 3)".

L203: or y1=y2

Yes! thank you for pointing this out.

Fig3A: but doesn't this figure show all the data, not just the night-time, wet conditions where we assume zero flow?

Yes, it is correct that the figure 3A includes all the data, but the baseline for the whole dataset should have been close to 1 and not somewhere between 1.4 and 1.5 as seen here.

L256: I don't understand this part. Please explain. why are these values eliminated?

Because the $\frac{v_1}{v_2}$ is an average of multiple instantaneous HPRs, the slope of each HPR will demonstrate how close they are to zero, which is what the slope should be if the sensors are perfectly symmetrical.

In the text it's explained accordingly (L183-186):

"The magnitude of the threshold chosen for the slope was taken from a modelled output of $\frac{v_1}{v_2}$ ratios performed by Burgess et al. (2001), were low sap flow velocities (5 cm h-1) combined with a small wound width (0.17 cm) were shown to display a slope of 0.001."

Your axis labels only show 0.01, we need more digits here!

Corrected, thank you.

L260: Is this a week were you assume that we have zero flow at night? Then state that.

No, it is not. I it is a zoomed in panel of a normal week of data. We included:
"Zoomed in panel of HPR data for one week of measurements to better see the diurnal pattern".

L265: You should show this in the plot - give the median and clearly indicate the areas on the plot with the erroneous data.

The read lines indicate the erroneous data. All data below or below these lines are filtered out.

L265: "Red line indicates threshold used for the quality control for this particular sensor. All data above or below these lines are filtered out as they did not correspond to HPR with |slope – median (slope)| < 0.003 s-1. Median slope for this sensor was -0.004"

Table1 and figure 6: Why not give the dates here? Or month and year?

We included the dates in the table description. Adding it directly to the header of the table would alter its format. Same for figure 6.

what do you mean by this?

L333-L336: "Because the HPR is an average of multiple instantaneous temperature ratios, it is difficult to ensure its accuracy only by visual inspection of the averaged output, without further statistical information. For each heat pulse measurement sequence, we suggest RSD and slope from linear regression of the instantaneous ratios versus time, to filter out random ratios and to ensure the quality of the data"

Combined explanations:

L337: What do you mean by this? erroneous? uncertain? Limited is too vague. L338: Not clear. This sentence is incomplete.

We have specified in the text accordingly:

"The HRM only measure values below 45 cm h-1, (Forster, 2017), due to a maximum ratio of v1/v2 = 20, for when it can be assumed accurately (Burgess et al., 2001; Marshall, 1958). Because our dataset showed no higher velocities than 8 cm h-1, it was not initially considered a limitation for the measurements.".

L347: S3 corrected to Fig. S3.

L384-L385: Deleted.

Please find the marked-up manuscript version below.

[revised manuscript text omitted]

[Figure]

Figure 2. Sensor placements, from top: thermocouple, heater, thermocouple, in an Aleppo pine. The x-axis represents the vertical distance from the heater, while the y-axis accounts for horizontal probe alignment from the heater. The axes should cross where the heater is inserted but was moved for a clearer view of the coordinates.

If probes are installed symmetrically, horizontally and vertically, in line with the heater, $x = x_1 = -x_2$ and $y_1 = -y_2 = 0$, equation 2 simplifies into a function that is not dependent on time:

$$V = \phi \ln\left(\frac{v_1}{v_2}\right)$$

$$\phi = \frac{K}{x}$$

[3]

where $\phi$ is a priori, a constant only depending on the placement of the probes and on the thermal diffusivity of both the xylem and the material used in the sensors, and x, a positive magnitude representing the distance from the heater element for the two probes

The $\frac{v_1}{v_2}$ is an average of multiple instantaneous heat pulse ratios (HPR), one for each second the temperature was registered for, from 60 to 100 seconds after the heat pulse release. The "perfect symmetry" assumption renders that the heat pulse ratio (HPR) remains constant with time if heat pulse velocity ($V$), thermal diffusivity ($K$) and probe positions (in both, x and y directions) have negligible variations during the time following each heat pulse (Marshall, 1958). Burgess et al. (2001) further demonstrated how empirical results initially differed from the ideal approach described by equation 3 due to the blocking of xylem vessels and probe misplacement. However, the study concludes that the heat pulse ratio converges asymptotically at least 60 seconds after the heat pulse release and, for at least, 40 seconds more (until 100 seconds after the heat pulse release), when ratio should be measured because the stable conditions. Our study argues that a visual inspection of heat pulse velocities ($V$ in equations 2 and 3) does not necessarily give enough information to decide if measured values are a good representation of the sap flow. The method does not consider that random ratios can arise, which due to the sensitivity of the method, are likely to occur in practice. On these premises, we have built a methodology to quality check sap flow measurements systematically by introducing a statistical analysis performed on the

instantaneous heat pulse ratio, acquired between 60 and 100 seconds after the heat pulse release. Hereafter, we will denote the averaged heat pulse ratio between 60 and 100 seconds as HPR. The quality check consisted of establishing threshold values for the slope of HPR against time and relative standard deviation (RSD), statistically defined as the standard deviation divided by the mean., and the slope of the HPR. The statistical information obtained would account for any deterioration of the measurement. Burgess et al. (2001) proposed two separately methods to correct for wounding and misalignment. The methods assume that errors arising from the wound inflicted by a sensor probe can be estimated using an empirical factor, whereas a misalignment of the probe needs to be calculated in situ. We propose a development of the misalignment correction method, while arguing that a statistical check of the HPR values would detect a deterioration of the signal caused by a worsening of the wound. This would lead to a smaller sample mean and hence a higher RSD and was The RSD was therefore chosen as a quality-check parameter along with the slope, which was a parameter proposed by Burgess et al. (2001).

**3.4.1 Logging specifications**

[revised manuscript text omitted]

---

## Author Response (AR4)

We thank the editor for the feedback and hope we have clarified any questions. Below point-by-point answers to each comment. Editors comments in black with the response from the authors in blue.

Is there data on this or is this just an assumption? Make this clear in the text.

Yes, it is an assumption. We modified the text to make this point clear.

"To establish a criterion that unified the effect of the hillslope, all the sensors were drilled into the uphill side of each tree trunk. It was assumed to offer a greater consistency regarding soil water retention, possibly higher at this orientation."

It is still not clear what exactly you mean by heat pulse ratio. The reader is trying hard to figure out: What is the difference to the heat pulse velocity that you describe in this paragraph?

We clarified:

"Marshall (1958) parameterised the instantaneous heat pulse velocity (V) in the HRM as a function of time (s) following a heat pulse, and an instantaneous heat pulse ratio (HPR), defined as HPR $= \frac{v_1}{v_2}$, where $v_1$ and $v_2$ are the downstream and upstream temperature increases, respectively, measured after the release of a heat pulse:

$$V = \frac{4Kt\ln\left(\frac{v_1}{v_2}\right) - (x_2^2 + y_2^2) + (x_1^2 + y_1^2)}{2t(x_1 - x_2)} \qquad [2]$$

where $K$ is the thermal diffusivity (cm$^2$ s$^{-1}$), $t$ is the time passed from the release of thermal pulse in seconds, $(x_1, y_1)$ and $(x_2, y_2)$ are the relative positions (cm) of the thermocouples to the heater (considering x-axis along the xylem and y-axis in the perpendicular direction) (Fig. 2).

I asked you before to clarify which sensor is below and which one above the heater. Downstream suggests that this is the sensor above the heater and upstream that this is the sensor below the heater (as the water is flowing upwards). However, your indices v1 and v2 suggest that v1 should be below the heater (lower on your vertical x-axis).

Yes, the downstream (v1) is the probe above the heater and the upstream (v2) is the probe below the heater. We have reformulated the caption of fig 2 to hopefully make it clearer:

"Sensor placements, from top: downstream thermocouple, heater, upstream thermocouple, in an Aleppo pine. The x-axis represents sap flow direction and the vertical distance from the heater, while the y-axis accounts for the horizontal probe distance from the x-axis. The axes should cross where the heater is inserted but are displaced for a clearer view of the coordinates".

This is confusing. why horizontally and vertically? We really need your language to be as precise as possible.
I suggest: If probes are installed equidistant to the heater in a vertical line , x1=-x2=x and y...

To allow for the y1 and y2 to be eliminated from the equation 2. However, we have rephrased:

"If probes are installed equidistant to the heater and aligned along the vertical axis, then $x = x_1 = -x_2$ and $y_1 = -y_2 = 0$, equation 2 simplifies into a function that is independent of time:

$$V = \phi \ln \left(\frac{v_1}{v_2}\right) \qquad\qquad [3]$$
$$\phi = \frac{K}{x}$$

where $\phi$ is a priori, a constant only depending on the placement of the probes and on the thermal diffusivity of both the xylem and the material used in the sensors, and x, a positive magnitude representing the distance from the heater element for the two probes.

For a whole sequence of instantaneous measurements, typically being one per second from the 60th second to the 100th second after a heat pulse release, the above equation should provide a heat pulse velocity slightly dependent on time as provided by Burgess et al. (2001), when wound width and sap velocities are small, with some inherent departures being explained only by instrumental errors."

Incomplete - no verb

Rephrased to:

"However, the study concludes that the HPR converges asymptotically to a slightly tilted straight line, at least 60 seconds after the heat pulse release and, for at least 40 seconds more (until 100 seconds after the heat pulse release), which is when the instantaneous heat pulse ratio should be measured because the temperature stabilises at this point".

I asked you in the previous document what you mean by random ratios - do you mean measurement errors? Then say that.

We refer to outliers in the heat pulse ratio dataset, but we have changed it to "measurement errors" to clarify.

The HPR should be constant during this period and thus the slope=0.

The HPR should be close to constant during this period, with a small slope.

So it is okay if the slope is large as long as it is always large? Sorry, I am still confused about this.

Sorry, it should be that the |slope – median (slope)| > 0.003 s$^{-1}$ were removed, which means we are filtering out measurements with slopes that deviates too largely from the median (outliers). The slope could be acceptably larger if the velocity was higher and the wound bigger as in Burges et al:

"All HPR with relative standard deviation > 5% and a |slope – median (slope)| > 0.003 $s^{-1}$ were removed to filter out measurements with large variability in the slope, i.e. with large deviation from the ideal slope at this velocity."

But this threshold is about the deviation of one period's slope from the median slope (across the entire data set), not about the slope itself - correct?

Yes, to filter out any measurement with too large deviation from the ideal slope, see comment below.

Is this HPR? According to your definition above HPR is the average v1/v2 for the 40 second period.

Yes, we have changed it in the text for consistency.

what do you mean by this?

Solar time replaced by Coordinated Universal Time (UTC).

Does this mean long monitoring periods? and
Monitoring period? (I would say that a campaign is when you are actually out there, doing experiments or taking manual measurements or installing equipment):

Yes, rephrased to:

If several calculations of $x_1$ and $x_2$ (equation 6), are performed during long monitoring periods, the non-intrusive approach of zero flow allows the parameterisation of misalignment as a function of time (equation 7).

This is still not clear, see my comment above. if you subtract the median slope, you just ensure that you have little variability in the slopes, but not that they are close to zero. Maybe I misunderstand things here - but this still means that you need to make it more clear

Yes, rephrased to:

Because the HRM is built on the theoretical assertion that the instantaneous HPR is closely linear with time and slightly tilted, specifically between 60 and

100 seconds after the release of a heat pulse, the slope of the HPR should be small, although dependent on the sap velocity and wound width (Burgess et al. 2001). Our HPR dataset displayed slope values close to what Burgess et al. (2001) proposed, but there were also measurements where the slope varied substantially, and we decided to filter out HPR with |slope – median (slope)| > 0.003 s$^{-1}$. This corresponded to 12% of the original dataset (Fig. 3C).

This needs to be explained somewhere. Are these erratic measurements?

Same as commented above, changed to: erratic measurements

Does this refer to assumed or to the ratio? Make sure if you need adverb or adjective here.

Rephrased to:
The HRM only measures values below 45 cm h$^{-1}$, (Forster, 2017), due to a maximum ratio of $\frac{v_1}{v_2}$ = 20, for when the ratio can be assumed to be accurate (Burgess et al., 2001; Marshall, 1958).

Changes in the misplacement? Or changes in the placement? Or changes in the alignment? It is a bit confusing here that you have misalignment on the one hand and misplacement on the other. Explain what the difference is. If there is no difference pick one term and stick to it.

Reformulated to:
"Even though Burgess et al. (2001) elaborated a correction method for probe misalignment we saw that changes in probe placement were detectable after each season."

I already asked this in the previous version: Is this placement or alignment?

We keep the term placement in this sentence.

[revised manuscript text omitted]

The $\frac{v_1}{v_2}$ is an average of multiple instantaneous heat pulse ratios, measuredone for each second the temperature was registered for, from 60 to 100 seconds after the heat pulse release.